# How well do Earth System Models reproduce the observed aerosol response to rapid emission reductions? A COVID-19 case study

Ruth A. R. Digby[1,2], Nathan P. Gillett[1], Adam H. Monahan[2], Knut von Salzen[1], Antonis Gkikas[3,4], Qianqian Song[5], and Zhibo Zhang[6,7]

[1]Canadian Centre for Climate Modelling and Analysis, Environment and Climate Change Canada
[2]School of Earth and Ocean Sciences, University of Victoria, Victoria, British Columbia, Canada
[3]Institute for Astronomy, Astrophysics, Space Applications and Remote Sensing, National Observatory of Athens (IAASARS/NOA), Athens Greece
[4]Research Centre for Atmospheric Physics and Climatology, Academy of Athens, Athens, Greece
[5]Atmospheric and Oceanic Sciences Program, Princeton University, Princeton, NJ, USA
[6]Physics Department, University of Maryland, Baltimore County, Baltimore, Maryland, USA
[7]Goddard Earth Sciences Technology and Research II, University of Maryland Baltimore County, Baltimore, MD 21250, USA

**Correspondence:** Ruth Digby (ruth.digby@ec.gc.ca)

**Abstract.**

The Spring 2020 COVID-19 lockdowns led to a rapid reduction in aerosol and aerosol precursor emissions. These emission reductions provide a unique opportunity for model evaluation, and to assess the potential efficacy of future emission control measures. We investigate changes in observed regional aerosol optical depth (AOD) during the COVID-19 lockdowns, and use these observed anomalies to evaluate Earth System Model simulations forced with COVID-19-like reductions in aerosols and greenhouse gases. Most anthropogenic source regions do not exhibit statistically significant changes in satellite retrievals of total or dust-subtracted AOD, despite the dramatic economic and lifestyle changes associated with the pandemic. Of the regions considered, only India exhibits an AOD anomaly that exceeds internal variability. Earth System Models reproduce the observed responses reasonably well over India but initially appear to overestimate the magnitude of response in East China and when averaging over the Northern Hemisphere (0°N-70°N) as a whole. We conduct a series of sensitivity tests to systematically assess the contributions of internal variability, model input uncertainty, and observational sampling to the aerosol signal, and demonstrate that the discrepancies between observed and simulated AOD can be partially resolved through the use of an updated emissions inventory. The discrepancies can also be explained in part by characteristics of the observational datasets. Overall our results suggest that current Earth System Models have potential to accurately capture the effects of future emission reductions.

## 1   Introduction

The emergence and rapid spread of COVID-19 in early 2020 had profound impacts on both individual behaviour and the large-scale economy. Following the detection of the first cases in Wuhan, China in December 2019 (World Health Organization,

2020a), and its categorization as a global pandemic in March 2020 (World Health Organization, 2020b), countries around the world began to implement a range of public health measures to limit the virus' spread. In many regions these included restrictions on public and private gatherings, domestic or international travel bans, curfews, and in some cases, stay-at-home orders.

One consequence of these dramatic economic and lifestyle changes was a rapid reduction in the emissions of aerosols and their precursors. This reduction has been widely reported on, with the majority of studies falling into one of two categories.

The first, more extensive category consists of studies investigating changes in local (predominantly urban) air quality during the pandemic. As summarized by Gkatzelis et al. (2021), more than 200 papers on air quality changes were published in the first 7 months of 2020 alone. These papers reported substantial reductions in major pollutants including $NO_2$, $NO_x$, $O_3$, $SO_2$, and particulate matter in cities around the globe. The spread in estimates varied dramatically within and between geographic regions. The intra-regional spread can be attributed in part to city- or nation-scale heterogeneity and in part to methodological differences between studies. Notably, Gkatzelis et al. (2021) found that approximately two thirds of these works did not account for interannual or in some cases even seasonal variability in aerosol levels; the importance of accounting for meteorological changes is emphasized in several recent works (Deroubaix et al., 2021; Diamond and Wood, 2020; Goldberg et al., 2020; Ordóñez et al., 2020). There have, however, been many rigorous studies of air quality changes (Chang et al., 2020; Jia et al., 2021; Ordóñez et al., 2020; Siciliano et al., 2020; Venter et al., 2020; Al-Abadleh et al., 2021; Liu et al., 2021; Sokhi et al., 2021; Wyche et al., 2021; Zhou et al., 2022; Kong et al., 2023).

The second category of COVID-19-based aerosol studies consists of model simulations of regional or global climate responses to this reduction in emissions. These studies, which include both idealized and proxy-based emission reduction scenarios for 2020, have reported weak to negligible climate responses, with top of atmosphere radiation changes comparable to interannual variability and changes in temperature or precipitation unlikely to be detectable in observations (Fasullo et al., 2021; Fiedler et al., 2021; Forster et al., 2020; Fyfe et al., 2021; Gettelman et al., 2021; Jones et al., 2021; Weber et al., 2020). These studies generally report reductions in zonally averaged aerosol optical depth (AOD) at $\sim$30°N in March 2020, and small or non-detectable changes in global aerosol optical depth. The magnitudes of these reductions are model-dependent and in some cases only detectable in the ensemble mean.

Few studies have bridged these categories, comparing the observed and simulated responses to the COVID-19 aerosol emission reductions. Among those studies that have incorporated both observations and simulations, the simulations have generally been used to help explain the observed changes by predicting control conditions in the absence of a COVID-19 perturbation (Goldberg et al., 2020; Griffin et al., 2020; Le et al., 2020; Hammer et al., 2021; Huang et al., 2021; Mashayekhi et al., 2021; Li et al., 2022). No studies have yet leveraged the observed aerosol response to the COVID-19 lockdowns for model evaluation purposes.

Reliable model simulations of the direct and indirect responses to a reduction in aerosol emissions are crucial for policy development. The Shared Socioeconomic Pathways (SSPs; Riahi et al., 2017) and Representative Concentration Pathways (RCPs; van Vuuren et al., 2011; Lamarque et al., 2011) all include a reduction in aerosol emissions over the coming decades. Because aerosols are often co-emitted with greenhouse gases (GHGs), some of these reductions will be the result of targeted

air quality legislation and some will be a side effect of climate policies. Depending on the sectors targeted by these policy measures, and the resulting cocktail of aerosol species that are affected, such reductions may lead to climate co-benefits or climate penalties (von Salzen et al., 2022). These uncertainties are compounded by the fact that aerosol-climate interactions remain one of the largest sources of uncertainty in future climate projections (Szopa et al., 2021). Predictions of future air quality changes are also highly uncertain (Szopa et al., 2021). The opportunity for model evaluation afforded by the COVID-19 pandemic is thus invaluable. In particular, the COVID-19 pandemic demonstrates the impacts of a rapid emission reduction, whereas previous observational constraints have been based on slower (e.g. decadal) emission trends (e.g. Ramachandran et al., 2022; Lund et al., 2023). There are both practical and scientific motivations for studying a rapid emission reduction. On the practical front, a short-but-strong signal is easier to disentangle from other sources of variability; we have continuous satellite observations that cover the entire study period; and the period is short enough the instrument drift is unlikely to be a concern. Scientifically, model simulations indicate that the presence and severity of potential climate penalties, including changes in mean and extreme temperatures and precipitation, may be proportional to the rate at which emissions are reduced (Acosta Navarro et al., 2017; Hienola et al., 2018; Samset et al., 2020; Shindell and Smith, 2019; Shindell et al., 2012; Sillmann et al., 2013). Although we do not investigate the climate response to COVID-19 in this work, understanding the aerosol response itself is an important first step.

In this work we use the observed AOD anomalies from Spring 2020 to evaluate model simulations of the response to a COVID-19-like reduction in aerosol emissions. Our work addresses two questions. First, how much did aerosol optical depth change in response to the dramatic lifestyle changes brought about by the COVID-19 public health measures? Second, how well do our models reproduce this response?

Both the air quality and climate impacts of a reduction in emissions depend on the response of atmospheric aerosol concentrations to emission changes, which is in general a complex and nonlinear dependence (Szopa et al., 2021; Kroll et al., 2020). The climate effect further depends on the resulting changes in scattering and absorption, which can be quantified in terms of aerosol optical depth. In this work we consider changes in AOD, rather than concentration, since it is more readily available from both model simulations and remotely sensed observations.

The models used in this work are taken from the COVID-19 Model Intercomparison Project (CovidMIP; Jones et al., 2021), which was developed to investigate the effects of a COVID-19-like reduction in aerosols and greenhouse gases. Although Jones et al. (2021) present an initial analysis of changes in aerosol optical depth, their primary foci were the radiative and climatic responses to the COVID-19 perturbation, and the drivers of the simulated aerosol changes were not investigated. Our analysis provides the first detailed investigation of aerosol changes in the CovidMIP models, as well as the first comparison between observed and CovidMIP-simulated changes.

In order to disentangle the signal of COVID-19 from other sources of AOD variability and model-observation discrepancy, we first derive a basic analytic framework consisting of three assessment metrics, and then systematically assess the influence of potential confounding factors on our results. This assessment is done by either correcting for these factors directly or conducting sensitivity tests to quantify their impacts on our metrics. The factors we assess are discussed in Section 2. Our datasets are described in Section 3 and the metrics by which we assess them are described in Section 4. Section 5.1 presents

the results of our basic analysis, and Section 5.2 describes our sensitivity tests. The implications of these results are discussed in Section 6.

## 2 Components of the AOD signal

Disentangling the signal of COVID-19 emission reductions from other sources of variability in an observed AOD field is a complex and nuanced task. Conducting robust comparisons between observed and simulated responses is yet more involved. We begin by highlighting the major considerations that need to be addressed in an analysis of this type: sources of AOD variability; factors that contribute to discrepancies between simulated and observed AOD fields, no matter the quality of the atmospheric model or satellite retrieval; and finally, the impacts of observational uncertainty.

### 2.1 AOD Variability

Aerosol optical depth is determined by natural and anthropogenic emissions, the chemical production of secondary aerosol species in the atmosphere, transport, and microphysical processes. All of these processes are affected to various degrees by meteorological conditions. Collectively, physical and chemical processes determine not only the distribution of atmospheric aerosols, but the optical depth that results from this aerosol burden.

Because these processes are highly interdependent, we simplify matters by grouping the drivers of AOD variability into three categories: variability in anthropogenic emissions, natural emissions, and meteorological conditions. Anthropogenic emissions vary over different timescales, including diurnal (e.g. weekday/weekend cycles of traffic emissions), seasonal (e.g. with changing heating/cooling requirements) and annual to decadal (e.g. due to socioeconomic changes and direct legislation). Natural emissions also differ substantially from one year to the next: mineral dust and sea salt emissions are strongly influenced by surface wind stress, and biomass burning emissions vary with storm activity (wildfire ignition by lightning) and aridity (soil moisture content, fuel availability). Once aerosols are in the atmosphere, their transport is affected by atmospheric circulation, and their removal by precipitation and dry deposition depends on hydrometeorological conditions, atmospheric stability, and winds.

The above processes determine aerosol burden. The AOD that results from this burden depends on the intrinsic characteristics of the aerosol (e.g. refractive index and particle morphology, which can change in response to variations in emission source, and whether components are internally or externally mixed) and on the ambient conditions (e.g. via hygroscopic growth). Because these microphysical characteristics ultimately depend on the changes in emission and meteorological processes listed above, we will consider them under the umbrella of those categories.

The observed AOD anomaly from Spring 2020 will combine these different factors, with the reduction in anthropogenic emissions due to COVID-19 making up an unknown fraction of the total. Our analysis must therefore account for these different sources of variability when assessing both the observed and simulated AOD signals. In the following sections, we describe first the differences that would be expected even if both models and observations were perfectly accurate, and then the impacts of observational uncertainty.

## 2.2 Differences between observed and simulated AOD in the absence of model error or observational uncertainty

Even given a hypothetical model that perfectly simulated atmospheric aerosol processes, and perfectly accurate satellite retrievals, differences would still arise between the observed and simulated AOD fields. These differences can be grouped into three main categories. First, a freely running model would produce a different realization of meteorological conditions than occurred in the real world, and so aerosols would be subject to different emission, transport, and depositional processes. Second, any errors in the model inputs (e.g. in the size of perturbation applied to represent COVID-19) would translate into biased simulations. Finally, the simulated and observed AODs would be recorded with different spatiotemporal sampling. Before any differences between the observed and simulated responses to COVID-19 can be attributed to model biases, then, these factors must be accounted for. (We discuss the role of observational uncertainty separately, in Section 2.3.)

We address AOD differences stemming from the differences in the observed and simulated realizations of the climate system in two ways. First, we compare AOD anomalies during COVID-19 to the scale of internal variability, both by looking at observed interannual variability over the reference period and by running ensembles of simulations to sample the models' internal variability. If an AOD anomaly is well outside of typical interannual variability, then it is likely to have been caused by more than just meteorological conditions. Equivalently, if the observed AOD change is well outside of typical interannual variability but the simulated AOD change is not, or vice versa, then the difference between these two responses is unlikely to be caused by differences in the simulated and observed meteorological conditions. Second, we can run simulations nudged to meteorological fields taken from reanalyses to constrain the simulated meteorological realization to be close to observations.

AOD differences arising from biases in the model input can be assessed by running simulations where the input emission inventories are varied. In this work we assess the sensitivity of our results to uncertainties in both the reference and COVID-19-perturbed aerosol emissions in one model, and discuss the extension of these results to the other models in our sample.

AOD differences can also arise from discrepancies between the spatial and temporal sampling of the different datasets. A simulated monthly-mean AOD value is the mean over all times of day and night, in all weather conditions. A satellite's monthly-mean AOD retrieval is the mean of AOD values collected at the satellite's particular overpass time, in clear-sky conditions (if it is a passive sensor), and at times when the retrieval was successful and not prevented by a myriad of potential limitations such as sun glint or complex terrain. We conduct a detailed assessment of these sampling differences (Supplementary Material S1) and demonstrate that in our data, the discrepancies caused by sampling differences are much smaller than those due to systematic biases between observed data products.

## 2.3 Observational Uncertainty

The three categories of difference discussed in Section 2.2 would occur even if both the model and satellite were perfect. However, satellite retrievals are characterized by both systematic and random measurement errors (Povey and Grainger, 2015; Sayer et al., 2020).

Estimation of retrieval biases and uncertainties is further complicated when considering quantities averaged in space and time. The degree to which individual retrieval errors are reduced by averaging depends on the relative contributions of random

and systematic components, which will vary both spatially and temporally (e.g. Povey and Grainger, 2015; Young et al., 2018). Even if the uncertainty was due entirely to random error, the effects of averaging would not be straightforward, since there is no reason to expect that the retrievals would be independent or identically distributed. Further complication arises when considering dust-subtracted aerosol optical depths, since the task of discriminating between aerosol species is separate from the determination of total AOD (Omar et al., 2009; Gkikas et al., 2021; Song et al., 2021), and it is not obvious how the respective uncertainties would be propagated even if they could be individually constrained. Given these limitations, spatial averages of remotely-sensed AOD are generally published without quantitative uncertainty estimates.

Considering the above complications, we do not provide uncertainty estimates for individual observational measurements. Instead we use the spread between observational products as an overall estimate of the uncertainty in the true AOD. As Supplementary Material S1 demonstrates, this spread is due primarily to systematic biases between products and not to differences in their spatiotemporal sampling. As Supplementary Material S1 further demonstrates, however, agreement between datasets is substantially improved when data are expressed as anomalies rather than absolute values.

## 3  Datasets

### 3.1  Model Simulations

The COVID-19 Model Intercomparison Project (CovidMIP; Jones et al., 2021) was developed to investigate the effects of a COVID-19-like reduction in aerosols and greenhouse gases. It includes contributions from twelve Earth System Models, all of which had previously participated in the 6th Phase of the Coupled Model Intercomparison Project (CMIP6; Eyring et al., 2016).

The CovidMIP scenarios, detailed in Forster et al. (2020), include the control scenario, a "two-year blip" experiment, and three long-term recovery scenarios. The two-year blip, which is the focus of this work, estimates aerosol and greenhouse gas emission reductions from mobility data (Forster et al., 2020; Le Quéré et al., 2020) for the period of Jan-June 2020; 66% of the June restrictions are then assumed to persist until the end of 2021, after which emissions recover linearly to the baseline. We limit our analysis to Jan-June 2020, for which the simulated emissions changes are based on observational proxies rather than projections. Further details on the calculation of these reductions, and an assessment of the sensitivity of our results to uncertainties in their estimation, are presented in Section 5.2.3.

The control scenario used in CovidMIP is SSP2-4.5 (Riahi et al., 2017), a middle-of-the-road scenario developed for the Scenario Model Intercomparison Project (ScenarioMIP, O'Neill et al., 2016) as part of CMIP6. The implications of this choice of reference are explored in Section 5.2.1. Throughout this work, SSP2-4.5 is stylized *ssp245* when we are referring to a model simulation experiment, and written as SSP2-4.5 when we are discussing the scenario more generally. The COVID-19-perturbed experiment is referred to as *ssp245-covid*. We additionally use the terms *reference*, *control*, and *perturbed* to refer to the 2015-2019 *ssp245*, 2020 *ssp245*, and 2020 *ssp245-covid* simulations respectively.

We use only those CovidMIP models for which aerosol concentrations were not prescribed, and for which data were available on the Earth System Grid Federation (ESGF; https://esgf-node.llnl.gov/search/cmip6/). For the control experiment *ssp245*, this

**Table 1.** CovidMIP models used in this work (in alphabetical order). "Published *od550dust*?" column indicates whether mineral dust optical depths were available on ESGF for each model.

| Model | Realizations | Published *od550dust*? | References |
|---|---|---|---|
| ACCESS-ESM1.5 | 30 | no | Ziehn et al. (2020) |
| CanESM5.0 | 50 | yes | Swart et al. (2019) |
| MIROC-ES2L | 30 | yes | Hajima et al. (2020) |
| MRI-ESM2.0 | 10 | yes | Yukimoto et al. (2019) |
| NorESM2-LM | 10 | yes | Seland et al. (2020) |
| UKESM1-0-LL | 16 | no | Sellar et al. (2019) |

requirement applies to the reference period (2015-2019) as well as to the 2020 experiment period. In addition, much of our analysis utilizes dust-subtracted optical depth, which is calculated by subtracting the aerosol optical depth due to dust (CMIP variable name *od550dust*) from the total AOD at 550nm (*od550aer*); however, *od550dust* is not published for all models. In total, four CovidMIP models met the data availability requirements for both total and dust aerosol optical depths, and an additional two models met the above criteria for total aerosol optical depth only. These six models, summarized in Table 1, sample the range of global AOD anomalies and climatic responses simulated by the full CovidMIP suite (Jones et al., 2021).

The most obvious outlier among these models is CanESM5.0, which exhibits an exceptionally large ensemble spread in AOD. This variability is caused by the production of spurious tropospheric dust storms, which have been attributed to errors in the tuning of mineral dust tracer parameters (Sigmond et al., 2023). Fortunately, only the dust tracers were affected, and the dust-subtracted AOD is consistent with that simulated by other CMIP6 models. This issue has since been corrected and will not be present in CanESM5.1. For the purposes of this analysis, CanESM5.0 is excluded from some analyses of total AOD.

Throughout this analysis, model results are described in terms of an ensemble median and 5th-to-95th percentile range. When results are presented as spatially averaged timeseries, the percentile range is calculated across the ensemble of spatially-averaged values.

### 3.2 Remotely Sensed Observations

This work incorporates remotely sensed observations of both total and mineral dust aerosol optical depths. The datasets used in our analysis are summarized in Table 2 and described in Sections 3.2.1 and 3.2.2 respectively.

### 3.2.1 Total Aerosol Optical Depth

Satellite observations of total AOD are taken from the Moderate Resolution Imaging Spectroradiometer (MODIS; King et al., 2013; Platnick et al., 2017), Multi-angle Spectral Radiometer (MISR; Diner et al., 1998), and the Cloud-Aerosol Lidar with Orthogonal Polarization (CALIOP; Winker et al., 2009). Level 3 gridded, monthly mean products are used from all three instruments. We additionally include an AOD estimate derived from CALIOP data by the Aerosol, Cloud, Radiation - Observation

**Table 2.** Observational datasets used in this work. Columns "AOD" and "DSAOD" indicate whether this dataset is used in our analyses of total and dust-subtracted optical depths respectively. "EOT:" Equatorial overpass time (local time). *: orbit changed in Sept 2018.

| Dataset Name | AOD | DSAOD | Instrument | Satellite | Type | Conditions | EOT | References and Notes |
|---|---|---|---|---|---|---|---|---|
| MODIS-Aqua | x | - | MODIS | Aqua | passive | cloud-free | 1:30pm | King et al. (2013); Platnick et al. (2017) |
| MODIS-Aqua-MIDAS | - | x | MODIS | Aqua | passive | cloud-free | 1:30pm | AOD from MODIS-Aqua; DOD from MIDAS (Gkikas et al., 2021, see Section 3.2.2) |
| MISR | x | - | MISR | Terra | passive | cloud-free | 10:30am | Diner et al. (1998) |
| CALIOP-AllSky-Night | x | x | CALIOP | CALIPSO | active | all-sky | 1:30am* | Winker et al. (2009) (CALIPSO official product) |
| ACROS-C | x | x | CALIOP | CALIPSO | active | cloud-free | 1:30am* | Song et al. (2021); Yu et al. (2015) (Developed by ACROS group) |

and Simulation (ACROS) group at the University of Maryland, Baltimore County (ACROS-C; Song et al., 2021), described further in Section 3.2.2.

There are two MODIS instruments, mounted on the Aqua and Terra satellites. We use MODIS Aqua data in our main analysis, due to an identified high bias in the MODIS Terra calibration (Levy et al., 2018), but compare with MODIS Terra data in Supplementary Material S1. The MISR instrument is mounted on Terra, and observes a smaller swath within the footprint of MODIS-Terra. As passive sensors, both MODIS and MISR observe only during the day and in cloud-free conditions. In contrast, CALIOP – onboard the Cloud-Aerosol Lidar and Infrared Pathfinder Satellite Observation (CALIPSO) platform – is an active lidar sensor and so can take measurements during the day or night. Because the absence of a solar background means that CALIOP's AOD has lower uncertainties at night (Young et al., 2018), we use the all-sky nighttime product in our main analysis. This product is compared with the cloud-free daytime product in Supplementary Material S1 to help quantify the effects of sampling differences between different datasets. Terra and Aqua have daytime equatorial overpass times of 10:30am and 1:30pm respectively. CALIPSO initially orbited in the same satellite constellation as Aqua, but in September 2018, moved to a lower orbit to maintain coincident measurements with CloudSat (ASDC, 2018).

AOD retrieved from MODIS is generally biased high relative to ground-based AERONET measurements (Wei et al., 2020; Levy et al., 2018), while AOD from MISR and CALIOP is generally biased low (Kahn et al., 2009; Schuster et al., 2012); AERONET is widely considered the gold standard for the evaluation of satellite AOD retrievals, despite limitations in its spatial representativity (Schutgens et al., 2020), because AERONET uncertainties are substantially lower than those from satellites. MODIS is also generally biased high, and MISR and CALIOP biased low, when comparing with other remotely sensed and

reanalysis datasets that are available for shorter periods of time (Vogel et al., 2022). As demonstrated in Supplementary Material S1, the spread in AOD estimates retrieved from these three datasets is dominated by systematic biases and not by sampling differences. Thus the spread between these datasets, which spans the range of AOD available from ground- and space-based instruments, provides a reasonable estimate of current observational uncertainty.

### 3.2.2 Dust Optical Depth

Much of this analysis considers dust-subtracted aerosol optical depths. Neither MODIS nor MISR produce a dust-specific product. However, other research groups have combined MODIS retrievals with auxiliary data to derive estimates of dust optical depth. In this work we use the ModIs Dust AeroSol (MIDAS; Gkikas et al., 2021) product, which combines total AOD from MODIS Aqua with an estimate of the dust-to-total ratio from the MERRA-2 reanalysis, to estimate the dust optical depth from MODIS Aqua. MIDAS also publishes a total optical depth estimate, which applies additional filters to the MODIS dataset. Our results are insensitive to the choice of MIDAS or MODIS total aerosol optical depth; we use MODIS Aqua in the analysis presented here.

CALIOP determines the composition of aerosol layers using a combination of retrieval information (depolarization ratio, integrated attenuated backscatter, layer height) and information about the underlying surface (Omar et al., 2009; Kim et al., 2018). The aerosol subtyping algorithm identifies seven tropospheric aerosol types, including dust, polluted dust, and dusty marine. However, these are separate classifications (e.g. "dust" refers only to clean dust, whereas "polluted dust" refers to dust mixed with urban pollution or biomass burning smoke) and there is no estimate of the total dust optical depth (Omar et al., 2009). We present results using the "dust" product (*AOD_Mean_Dust*) but acknowledge that it does not capture the entire contribution of dust to AOD, particularly in heavily polluted regions.

The ACROS-C dataset is included in our analysis to provide a secondary estimate of dust optical depth from CALIOP retrievals. ACROS-C uses the lidar depolarization ratio to distinguish dust by shape (Yu et al., 2015; Song et al., 2021). The ACROS group also publishes a MODIS-based dust optical depth product, but this dataset is not available for 2020 so is excluded from our analysis.

## 4 Methods: Quantifying the Impacts of COVID-19

In the Introduction, we presented two motivating questions for this work: First, do observations exhibit a detectable response to the COVID-19 emissions reductions? And second, how well do current models reproduce this response? Here we present the quantitative metrics by which these questions are addressed. Throughout, anomalies are calculated relative to the 2015-2019 March-April-May (MAM) mean.

For the models, we consider a COVID-19 response to be "statistically significant" if there is a statistically significant difference between the control and COVID-19-perturbed ensemble means in 2020, based on a two-sided Students t-test. Because the only difference between the control and perturbed ensembles is the absence or presence of a COVID-19 perturbation, we can compare the ensembles directly and avoid the challenge of disentangling year-to-year changes in the underlying anthropogenic

emissions from internal variability between realizations. Note, however, that this test is sensitive to ensemble size and that a larger ensemble will be more likely to yield a significant response.

Defining detectability for the observations is more challenging, because there is no "control observation" from which to estimate the AOD that would have been measured in 2020 had COVID-19 not occurred. It would not be sufficient to use the mean and variance calculated from the reference period as a control: the using the mean would neglect the impacts of underlying trends in the emissions, and a five-year reference period is too short to provide a robust estimate of the variance. Instead we borrow an approach from the field of detection and attribution (Eyring et al., 2021) and compare the ensemble of observed anomalies to a multimodel control ensemble (MMEc) constructed by randomly drawing an equal number of control simulation anomalies from each model. This comparison relies upon two assumptions: that the models realistically simulate internal variability, and that they capture any underlying trends in AOD over the reference period. These assumptions determine the spread and best estimate, respectively, predicted by the MMEc.

We assess the first of these assumptions by comparing observed and simulated variability over the reference period in our regions of interest. We calculate the variance of the region-mean AOD field over the reference period for each simulated ensemble member, the MMEc, and the observational datasets, and compare the spread in these estimates of variance. Although individual models may over- or underestimate the interannual variability in some regions, we cannot reject the null hypothesis that the MMEc and observations have the same interannual variability, based on a two-sided Welch's t-test. The only exception is for India, where the MMEc overestimates the variability in total and dust-subtracted aerosol optical depth; as a result, estimates of observational detectability will be conservative (i.e., anomalies are less likely to be found to be statistically significant). In a similar analysis of the variability over a longer baseline (2007-2019), using a subset of models for which these data were available, the total-AOD variability of the MMEc is consistent with that of the observations in all four regions.

Having determined that it is reasonable to treat the simulations and observations as having similar internal variability, we derive a statistical test with which to compare observed and simulated AOD anomalies. We use a modified Welch's t-test of the form

$$t = \frac{\bar{X}_1 - \bar{X}_2}{s_{\bar{\Delta}}} \qquad \text{where} \qquad s_{\bar{\Delta}} = \sqrt{\frac{s_1^2}{n_1} + \frac{s_2^2}{n_2}} \tag{1}$$

where $s_1^2$ and $s_2^2$ are unbiased estimators of the variance in the simulations and observations respectively. As outlined above we assume that the observations and simulations have the same variance due to internal variability, $s_1^2$, which we calculate from the simulated ensemble spread in the year under consideration. The observations additionally have a contribution to their variance due to observational uncertainty, $s_0^2$, calculated from the spread in observational estimates in that year. The total variance in the observations is then $s_2^2 = s_0^2 + s_1^2$. We use this t-test to determine whether the observed anomaly is statistically significant, through comparison with the MMEc, and to determine whether the individual models' simulated anomalies are consistent with the observed anomaly, by comparison with the control and perturbed experiments in turn. We present results for t-tests performed on region-mean values; using spatially-resolved comparisons adds little information and does not change our results.

The second assumption, that the models capture any trends in AOD over the reference period, may break down in some geographic regions. However, although some regions have nonstationary anthropogenic emissions over the 2015-2019 reference period, we do not detrend our datasets. With only a five-year baseline, the observed timeseries is heavily influenced by interannual variability which masks underlying trends in anthropogenic emission changes. With a longer baseline, the assumption of a linear trend would be unfounded; China, for example, had steadily-increasing anthropogenic aerosol emissions until their implementation of the Action Plan on the Prevention and Control of Air Pollution in 2013, since which emissions have been rapidly declining (Zheng et al., 2018). Given these limitations we work with anomalies calculated relative to the 2015-2019 mean but discuss the impacts of underlying trends on our results throughout.

## 5 Results

### 5.1 Comparison between observations and CovidMIP models

We begin with an assessment of the changes in total and dust-subtracted AOD during the COVID-19 lockdown periods. Four regions are selected for analysis: the Northern Hemisphere (restricted to 0-70°N), and three major anthropogenic aerosol source regions, namely East China (100-122°E, 20-40°N), India (70-90°E, 5-30°N), and Europe (10°W-35°E, 35-70°N). East China and India were defined following Wang et al. (2021), with the India domain extended south to include all of Sri Lanka, and the European domain was selected to maximize (minimize) the land (ocean) enclosed. The Northern Hemisphere domain was restricted to 0-70°N to enable comparison with observations. The boundaries of these regions are illustrated in Supplementary Figure S7.

The timing of the strictest lockdowns, as defined by the Oxford Stringency Index (Hale et al., 2021), varies between these regions but occurs for all between March and May of 2020. To aid in inter-region comparisons, and to increase the signal-to-noise ratio over what would be obtained from analyses of single months, we evaluate the MAM-mean anomaly in all four regions.

#### 5.1.1 Response of Total AOD to the COVID-19 Emission Reductions

The observed and simulated total AOD anomalies over 2015-2020 are summarized in Figure 1. Of the four regions assessed, only India exhibits a statistically significant observed response to the COVID-19 lockdowns. In East China and Europe, the observed anomalies are clearly within the range of interannual variability over the reference period. In the Northern Hemisphere, two of the four observational products suggest anomalously low AOD compared to the preceding 5 years, although this excursion is less clearly anomalous when considering the apparent negative trend in AOD; the other two data sources do not show particularly large anomalies.

Simulated results vary from model to model. In the Northern Hemisphere, two of the six models exhibit statistically significant differences between the control and perturbed anomalies, but in most cases, both the control and perturbed ensembles are consistent with the observed anomalies in this region. In East China, four of the six models exhibit a statistically significant

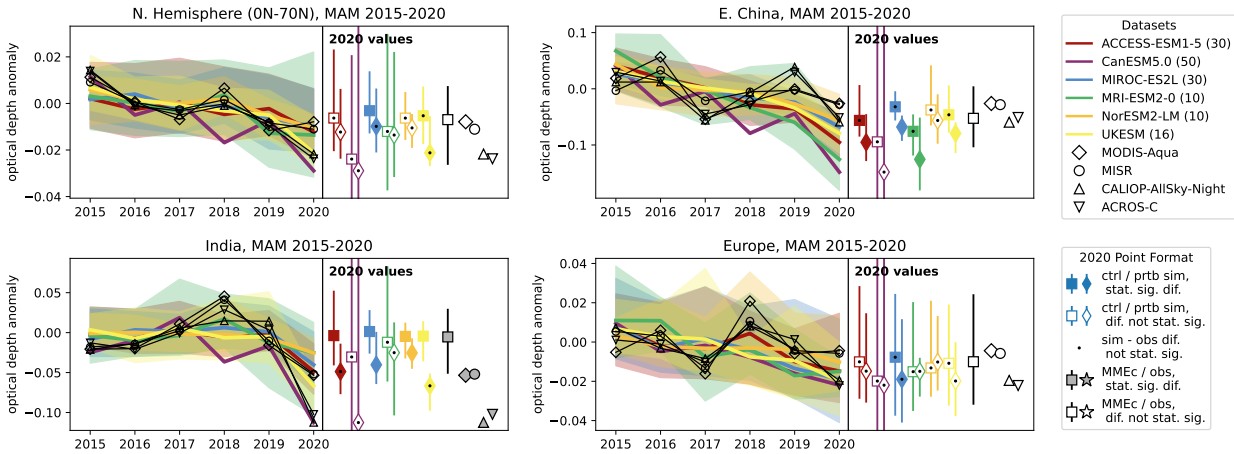

**Figure 1.** MAM-mean AOD anomalies in CovidMIP models (colour, ensemble sizes noted in parentheses) and in remotely-sensed observations (black), in the Northern Hemisphere and in 3 major source regions. The left-hand portion of each panel shows timeseries of AOD anomalies over 2015-2020, with datasets denoted as per the upper legend. For the simulations, lines and shaded envelopes indicate ensemble medians and 5th to 95th percentile ranges respectively. The right-hand portion of each panel plots the 2020 AOD values: two points for each model, corresponding to the control and COVID-19-perturbed simulations (square and diamond markers respectively, with error bars showing the 5-95% ensemble range), and one point for each observational data product (black outlines, no error bars). These points are formatted according to the results of our statistical comparisons, as summarized in the lower legend. In this legend, the blue-coloured and star-shaped markers represent simulated and observed datasets respectively, each of which is plotted with a different colour/marker as indicated in the upper legend. Model simulations that exhibit a statistically significant COVID-19 response (i.e., separation between control and perturbed ensembles) are plotted with filled symbols, and those that do not are plotted with open symbols. Black dots indicate simulated anomalies that are not significantly different from the ensemble of observed anomalies. The single, black-outlined square with error bars, plotted between the simulated and observed anomalies, shows the MMEc against which observations are compared. If there is statistically significant separation between the MMEc and observations (i.e., a detectable observed response), then both the MMEc and observations' markers have a solid fill. Note that the 4 panels have different vertical scales. Note also that because of the spuriously large AOD spread in CanESM5.0, only the ensemble median is shown in the timeseries; the 2020 ensemble ranges extend beyond the limits of the figure.

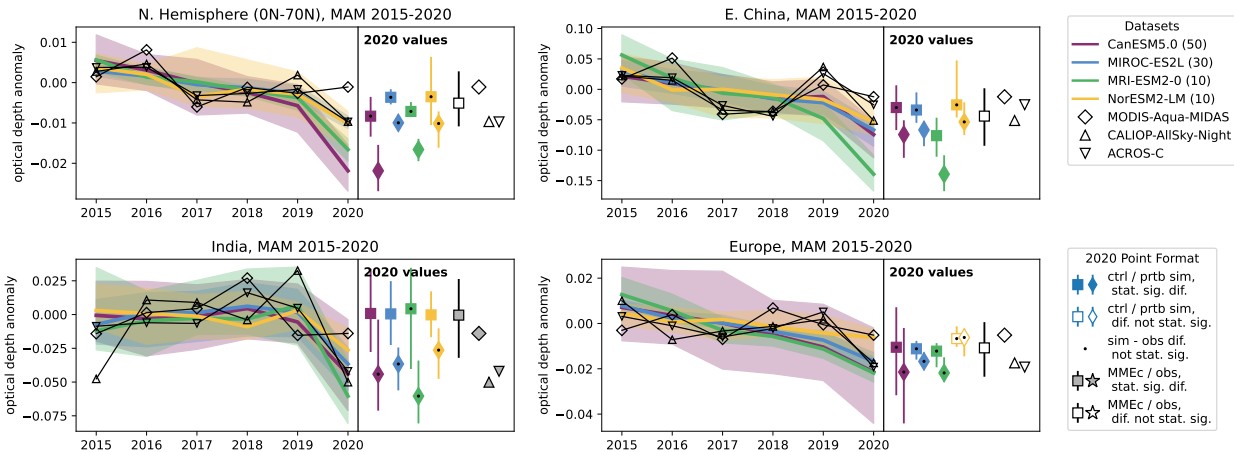

**Figure 2.** As Figure 1, but with the dust contribution removed from the optical depth. The ensemble spread is shown for CanESM5.0 here, unlike in Figure 1, since subtracting off the dust component removes the spuriously high AOD variability in this model.

COVID-19-perturbed anomaly, which in general appears to be over-estimated: in these models, there is a statistically significant between the observations and the perturbed ensemble, but not between the observations and the control. In India, like East China, four of the six models simulate statistically significant separation between the control and perturbed ensembles. Unlike in East China, however, these responses are in good agreement with the observed anomalies: four of the perturbed ensembles are consistent with the magnitude of the observed ensemble, whereas only two of the control ensembles are. Finally, there is also good agreement between the observed and simulated anomalies in Europe: only one model shows a statistically significant perturbation, and in all models, both the control and perturbed ensembles are consistent with the observed anomalies.

The existence of some disagreement between the observed and simulated responses is not unexpected. As outlined in Section 2, the total-AOD signal is influenced by many factors beyond the changes in anthropogenic emissions. The impacts of these factors are assessed in the following sections.

### 5.1.2 Response of Dust-Subtracted AOD to the COVID-19 Emission Reductions

We next investigate the AOD signal when the contribution from mineral dust has been removed (dust-subtracted AOD, or DSAOD). In our regions of interest, the variability in total aerosol optical depth is dominated by the variability in mineral dust, which was not directly impacted by COVID-19 lockdowns. Its presence may thus substantially mask any anthropogenically-driven AOD changes. Although dust emission could have been indirectly affected via, e.g., changes in temperature or precipitation induced by $CO_2$ or aerosol changes during the COVID-19 lockdowns, we expect these effects to have been small; as described above, simulations suggest only weak climate responses to the COVID-19 emission changes.

Figure 2 reproduces Figure 1 but for DSAOD, showing results from the four CovidMIP models for which dust optical depths were available. As with the total AOD, an observed response is only detectable in India. Accounting for the role of dust improves model-observation agreement in this region, with all four models assessed exhibiting significant COVID-19 responses that agree with the magnitude of the observed anomaly. This improvement is partially due to the fact that the observed dust optical depth was anomalously low in 2020, as seen in our results and as reported elsewhere (Smith et al., 2022; Wei et al., 2022). Because the simulations were allowed to evolve freely, they would not in general have reproduced the meteorological conditions that led to this anomaly.

Because we are using CALIOP's clean dust product, the DSAOD we calculate still contains contributions from dust that is mixed with pollution or marine aerosol. The negative 2020 anomaly observed in India by CALIOP may thus be due in part to the anomalously low dust optical depth noted above. If this is the case, the fact that the simulations are in qualitatively better agreement with the CALIOP anomaly than the MODIS anomaly might suggest that the simulations are overestimating the "true" DSAOD reduction. Reassuringly, however, the ACROS estimate is in good agreement with both the CALIOP estimate and the observations, despite using an independent method to estimate dust optical depth.

Good agreement between the observations and simulations is also found in Europe. Despite model-to-model differences in the DSAOD trend over the reference period, all models simulate both control and perturbed ensembles that are consistent in magnitude with the ensemble of observations. Although three of the four models simulate statistically significant perturbations, whereas the observations do not show a statistically significant response, the magnitudes of the simulated perturbations are very small (indeed, smaller than the spread in observed anomalies).

Models appear to overestimate the DSAOD response to COVID-19 in the East China and Northern Hemisphere domains. The overestimation is more ubiquitous in East China, where only NorESM2-LM simulates a COVID-19 perturbation consistent in magnitude with the observed anomaly. CanESM5.0 and MIROC-ES2L simulate somewhat larger (more negative) anomalies in this region. The fourth model, MRI-ESM2-0, exhibits substantially different behaviour than the others due to a much steeper trend through the reference period. When the Northern Hemisphere ($0°$N-$70°$N) is considered as a whole, two of the models simulate perturbations consistent with the observed anomaly, and two overestimate the AOD reduction. The spatial origins of these overestimations differ: in MRI-ESM2-0, the Northern Hemisphere-averaged anomaly is due almost entirely to the strong negative anomaly over Asia; in CanESM5.0, ensemble-median anomalies are negative throughout the entire region (Supplementary Figure S8).

Given the short reference period and substantial interannual variability of the observations, it is challenging to identify whether the simulated trends are representative of those observed. In East China there is some indication that the models may simulate marginally more negative trends than the observations (more visible in the raw data shown in Supplementary Figure S6, and when the 2020 anomaly is not included in the timeseries, and consistent with the results of Lund et al. (2023) for trends over 2005-2017), which could imply that the MMEc may inadequately represent the range of plausible "control observations." However, this discrepancy – if it exists – does not appear sufficient to explain the absence of a statistically significant anomaly in the observations. Visual inspection suggests that the observed 2020 anomalies are consistent with DSAOD excursions measured

over the preceding 5 years, even taking any potential trends into account, whereas the simulations show an obvious decrease in 2020. This behaviour is particularly clear when considering the raw data shown in Supplementary Figure S6.

## 5.2 Sensitivity Tests

Having established the magnitude of the observed optical depth anomaly in Section 5.1, and identified a number of discrepancies between this anomaly and those simulated by the CovidMIP models, we now conduct a series of sensitivity tests to probe the influence of potential confounding variables on the above results. These sensitivity tests are conducted using CanAM5 (Cole et al., 2023), the atmospheric component of CanESM5. Specifically we use CanAM5.1, a version of the model in which the spurious dust storms have been corrected through retuning of the hybrid tracer parameters (Sigmond et al., 2023). The complete details of these tests are provided in the supplementary material.

### 5.2.1 Sensitivity to Control Emissions

In CovidMIP, the COVID-19 perturbation was applied to the SSP2-4.5 baseline. This inventory was developed in the 2010s as a projection for the years 2015-2100 (starting from proposals by Krieler et al. (2012) and van Vuuren et al. (2012)), and has known differences from the true emissions that have occurred since its development. In particular, it did not account for recent clean air legislation in China, which has substantially reduced the emission of aerosols and their precursors since the early 2010s (e.g. Wang et al., 2021; Zheng and Unger, 2021; Paulot et al., 2018). Furthermore, biomass burning emissions in SSP2-4.5 are described by a linear projection which does not capture the high interannual variability of these sources.

We investigate the sensitivity of our results to biases in the baseline emissions inventories by constructing updated inventories for both anthropogenic and biomass-burning aerosol emissions, and using these inventories to run 10-member ensembles in CanAM5.1. Anthropogenic emissions are based on the 2021-04-21 release of the Community Emissions Data System (CEDS) emission inventory (Smith et al., 2021), which provides emissions estimates up to 2019, and year-specific biomass burning emissions are taken from the Global Fire Emissions Database (GFED v4.1s). Details of the inventory construction and model configuration are provided in Supplementary Material S3.

These simulations ("CanAM-new-emis") are compared to the original CanESM5.0 ensemble in Figure 3, which shows dust-subtracted aerosol optical depths as in Figure 2. In all regions except India, updating the baseline emissions inventory reduced the median separation between control and perturbed ensembles, because the COVID-19 perturbation was applied as a percent change to a smaller initial value (Supplementary Figures S9, S11). The effects of this reduction vary: in East China, the update is sufficient to bring the simulated COVID-19 signal into agreement with the observed anomaly (i.e., the separation between observed and simulated ensembles is no longer statistically significant), whereas in the Northern Hemisphere the simulated response is brought somewhat closer to the observations but the difference remains significant. In Europe the main effect of the update is to remove the simulated trend over the 2015-2019 reference period, bringing the simulations into qualitatively better agreement with the observations over this period and reducing the 2020 anomalies to which the trend had contributed. In India, emissions were largely unaffected by the update, so the results do not change except for the loss of the large-negative-anomaly tail on the perturbed distribution. This change could be due either to the updated inventory or to the reduced ensemble size.

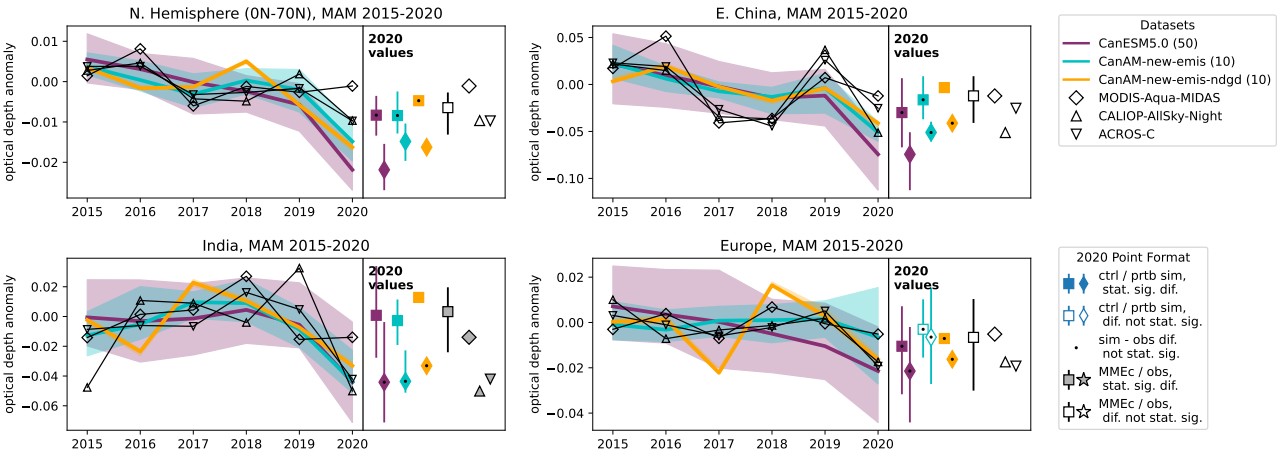

**Figure 3.** As Figure 2, but comparing dust-subtracted AOD in the 3 CCCma simulations: CanESM5.0 (the original CovidMIP implementation), CanAM-new-emis (an atmosphere-only simulation with updated anthropogenic and biomass-burning aerosol emissions; Section 5.2.1), and CanAM-new-emis-ndgd (as CanAM-new-emis, but nudged to meteorological fields taken from ERA5 reanalysis; Section 5.2.2). The MMEc in this figure is drawn from CanESM5.0, CanAM-new-emis, and CanAM-new-emis-ndgd; using the MMEc from Figure 2 does not change the statistical significance of the observed anomaly.

These results suggest that in India, inferences drawn from the other CovidMIP models will likely not be affected by biases in the underlying baseline inventory as long as the applied perturbation is realistic. In the Northern Hemisphere, East China, and Europe, the apparent overestimation of the COVID-19 response identified in CanESM5.0, MIROC-ES2L, and MRI-ESM2-0 may have been partially caused by overestimates of the control emissions and thus of the absolute magnitude of the COVID-19 disruption. Confirming this hypothesis is beyond the scope of this work. The changes in DSAOD magnitude through the reference period which occur as a result of the updated emissions are explored in Section 6 and Supplementary Material S3.

### 5.2.2 Sensitivity to Meteorological Conditions

Once aerosols have been emitted, the aerosol burden that remains in the atmosphere and the optical depth that this burden produces are determined by the meteorological conditions. We assess the degree to which meteorological conditions impacted the magnitude of the apparent COVID-19 response using a second CanAM5.1 experiment, CanAM-new-emis-ndgd.

CanAM-new-emis-ndgd simulations were nudged to temperature, wind, and humidity fields taken from the European Centre for Medium-Range Weather Forecasts (ECMWF) 5th Generation Reanalysis (ERA5; Hersbach et al., 2020); details of this nudging process are described in Supplementary Material S3. Anthropogenic and biomass burning aerosol emissions in CanAM-new-emis-ndgd were taken from the same updated inventories as were used in CanAM-new-emis.

A comparison between the absolute AOD simulated by CanESM5.0, CanAM-new-emis, and CanAM-new-emis-ndgd is provided in Supplementary Material S3. The degree to which updating the inventory and nudging towards ERA5 reanalysis improves the agreement of CanAM5.1 with the observed AOD fields (in terms of both average magnitude and the pattern of interannual variability) varies from region to region and, in the Northern Hemisphere region, depends on which satellite dataset is considered as reference. In general the nudged simulations exhibit higher interannual variability than the free-running simulations do, although this variability generally falls within the envelope of the larger coupled ensemble. In Europe, the nudged ensemble exhibits substantially higher interannual variability than do either of the free-running ensembles or the observations, indicating that the model may underpredict the variability of and AOD sensitivity to temperature, winds, and/or humidity in this region.

Results from CanAM-new-emis-ndgd are compared with CanAM-new-emis and CanESM5.0 in Figure 3. Because the ensemble spread in CanAM-new-emis-ndgd is negligible, we can assume that any difference between the control and perturbed ensembles is due to the signal of COVID-19 and not to internal variability. It is worth noting that the meteorological fields towards which the simulations were nudged may themselves have been impacted by the COVID-19 emission reductions. For instance, a reduction in scattering aerosols could have resulted in local warming. The control and perturbed CanAM-new-emis-ndgd simulations are therefore not fully independent. However, in all regions except Europe, the separation between the control and perturbed CanAM-new-emis-ndgd ensembles is large enough that this effect is unlikely to materially affect our conclusions.

The 2020 anomalies in the nudged simulations, considered in the context of the variability differences between datasets, are largely consistent with those from the free-running simulation CanAM-new-emis. In East China, both CanAM-new-emis-free and CanAM-new-emis-ndgd simulate lower interannual variability than do the observations; the pattern of variability is perhaps somewhat improved in the nudged ensemble. In the previous section we found that updating the emissions inventory substantially reduced the magnitude of the perturbed anomaly, bringing it into agreement with the observed anomaly. In that model configuration, both control and perturbed ensembles simulated 2020 anomalies that were consistent with the observations. Nudging results in a small positive shift in both the control and perturbed anomalies, such that the control moves just outside the range of the observed ensemble. However, since the difference between the control and the observations is substantially smaller than the spread in observational estimates, it is reasonable to describe the nudged control ensemble as still qualitatively agreeing with the observations.

In India, the nudged control ensemble exhibits a positive anomaly suggesting that meteorological conditions may in fact have partially masked the strength of the COVID-19 response. Because the observed interannual variability is reproduced less well in the nudged than the free-running simulations, detailed comparisons between the observed and simulated 2020 anomalies would not be robust; however, it is reassuring to note that the 2020 perturbed anomaly is in excellent agreement with the observations, and the results of our statistical comparisons remain unchanged.

In Europe the nudged ensemble exhibits higher interannual variability than do either of the free-running simulations. The pattern of positive and negative anomalies is consistent with, although substantially amplified relative to, observations from MODIS-Aqua; however, this pattern of variability is not reproduced in the two CALIOP-derived datasets. The simulated

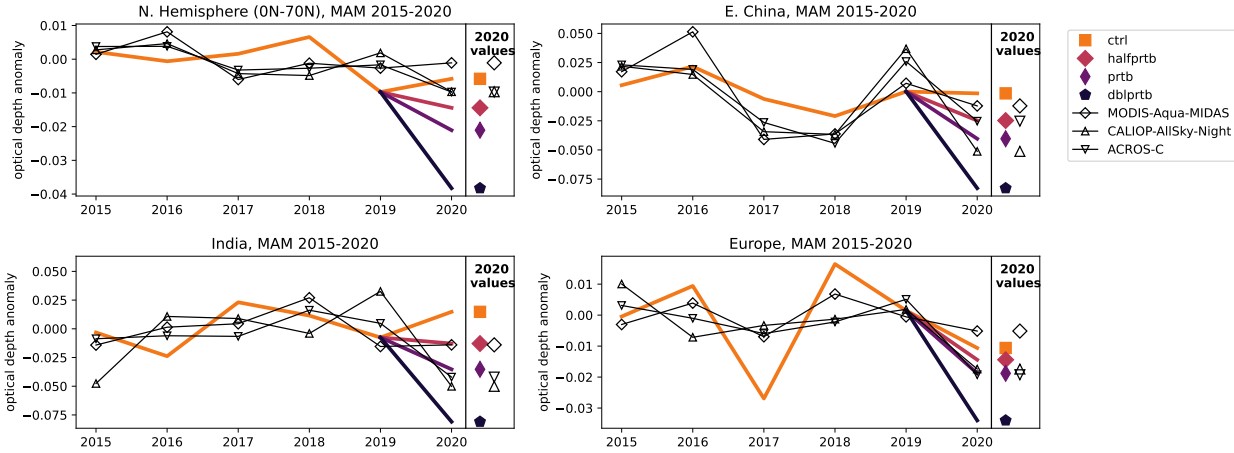

**Figure 4.** Effects of varying the magnitude of the simulated COVID-19 perturbation in CanAM5.1. The control simulation, nudged to ERA5 fields, is shown in orange. The remaining symbols show simulations forced with COVID-19 perturbations half, one, and two times the magnitude of the original CovidMIP perturbation. All simulations use the updated baseline inventory introduced in Section 5.2.1.

COVID-19 perturbation is small relative to this variability and, as in all previous ensembles, the control and perturbed 2020 anomalies are both consistent with the observed ensemble.

When averaged over the entire Northern Hemisphere (0°N-70°N), neither the free-running nor nudged ensembles reproduce well the observed pattern of interannual variability. Overall, our results here are unaffected by nudging: the control ensemble is consistent with the observed anomalies, and the perturbed ensemble values are too negative.

In all regions, the separation between the control and perturbed CanAM-new-emis-ndgd ensembles is on the order of the spread in observational estimates, suggesting that further observationally-based model evaluation may not be feasible given current observational uncertainties. Improving observational constraints beyond this point would require substantially reducing the differences between remotely sensed products retrieved using different instruments and algorithms.

### 5.2.3 Sensitivity to Perturbation Size

Finally, in this section we address the potential impact of uncertainties in the perturbation applied to simulate COVID-19. Although as stated above our observational constraints of model performance are limited by observational uncertainty, it is still informative to understand how sensitive our comparisons are to uncertainties in the perturbation size.

The aerosol emission reductions used in the simulations considered to this point were estimated from mobility data by Forster et al. (2020) and gridded for use in CovidMIP by Lamboll et al. (2021). In brief, the reductions were determined as follows. Google mobility data were used to estimate the country-by-country reduction in activity at transit stations and at residential, workplace, retail, and recreation locations, with Apple mobility data used as a secondary check. These reductions were then

used to estimate sector-by-sector $CO_2$ emission reductions following the methodology of Le Quéré et al. (2020) but using the mobility-data-derived estimates of activity changes in each sector. The original Le Quéré et al. (2020) estimates, derived from an estimate of "confinement level" on a scale of 1-3 based on government and media reporting, were used for countries for which mobility data were unavailable (notably for our analysis, this includes China) and for aviation and shipping sectors. Non-$CO_2$ emissions for 2020 were then determined by scaling each species' 2015 emissions by the ratio of $CO_2$ emissions in 2015 and 2020. This scale factor was determined uniquely for each country and sector.

This approach has the advantage of being self-consistent and bottom-up, but does come with a number of limitations. Notably, mobility data may not be a good proxy for emission changes in all sectors (e.g. some industries may have background emission even in the absence of active production), and changes in Apple or Google mobility data may overestimate the actual reduction in traffic (e.g. there may be a correlation between the subset of the population who have smartphones and those who have the opportunity to work from home, particularly in less-affluent nations). Gensheimer et al. (2021) measured differences of up to 60% between mobility data and local traffic changes, and could not determine a unique functional relationship between these quantities that applied in all regions. Finally, $CO_2$ and non-$CO_2$ species have not evolved in tandem at the country/sector level over 2015-2020 (Smith et al., 2021), so scaling all species by changes in $CO_2$ introduces further error to the analysis. There is thus substantial uncertainty in the CovidMIP emission reductions.

We perform a sensitivity test by running nudged, baseline-corrected CanAM5.1 simulations with a range of perturbation strengths: no perturbation (the control), and perturbations of half, one, and two times the original CovidMIP experiment. We simulate only a single realization of each because, as demonstrated in Figure 3, nudging effectively removes all ensemble spread in AOD.

Given the uncertainty in the observations, and the fact that nudging does not result in particularly good agreement with the observations (Supplementary Figures S11 and S12), it is not constructive to directly compare the magnitude of the observed and simulated AOD anomalies as a function of perturbation strength. However, comparing the spread in the observations to the spread in simulated responses to different perturbation strengths can indicate whether or not uncertainties in the perturbation would affect our model evaluation results (i.e. whether they would be visible over the observational uncertainty).

The results of these simulations are presented in Figure 4. In all regions, the spread in the observed anomalies is larger than the separation between the control and half-strength, or half- and full-strength, simulated perturbations. So although a doubling of the perturbation strength is clearly inconsistent with the observations, uncertainties of up to $\sim$50% would not impact our model evaluation assessment. In Europe, the perturbation strength could be very nearly doubled without the spread exceeding observational uncertainties. In conclusion, modest uncertainties in the perturbation size are unlikely to have impacted our results in the preceding sections.

## 6    Discussion

### 6.1    Summary and Synthesis

In this work we have used remotely-sensed observations of AOD changes from the spring of 2020 to evaluate the response of aerosol optical depth to emission changes in CMIP6-class models from the CovidMIP project. We then performed sensitivity tests in CanAM5, the atmospheric component of CanESM5, to assess the influence of model inputs and meteorological variability on our comparison.

Ａ The response of total aerosol optical depth to the COVID-19 emission reductions is, unsurprisingly, very small and not
statistically detectable in most regions. Aerosols are characterized by substantial interannual variability, many drivers of which were not affected by the COVID-19 lockdowns. Detection of a response is further hampered by biases in model inputs and by uncertainty in the observations.

When the mineral dust component has been removed from aerosol optical depth measurements, good agreement between the observed and simulated responses is found in India (where both exhibit a significant COVID-19 response) and in Europe
(where a difference can only be detected in the simulated ensemble means, and where both control and perturbed anomalies are consistent with the observations). Models appear to overestimate the COVID-19 response in the Northern Hemisphere generally and in East China specifically, although our results suggest that a more accurate emissions inventory can reduce this discrepancy. Our conclusions appear to be insensitive to modest uncertainties in the magnitude of the simulated COVID-19 perturbation, in part because of the large spread in observational estimates of the COVID-19 response. This spread, which is
caused by systematic biases between the different datasets, prohibits a more detailed evaluation of the simulated response.

The sensitivity studies presented here incorporate aerosol emission inventories which have been updated from the SSP2-4.5 scenario used in the original CovidMIP experiment. As well as altering the magnitude of the COVID-19 perturbation, these updates cause important changes in the magnitude of DSAOD simulated through the reference period. In Supplementary Material S3 we decompose simulated changes in the magnitude of DSAOD over 2015-2019 into reductions caused by changes
in anthropogenic emissions, biomass burning emissions, and model configuration (coupled vs atmosphere-only). In all regions, DSAOD was reduced in CanAM-free from the original CanESM5.0 CovidMIP simulation, but the relative contributions to this reduction varied. In East China, the reduction in anthropogenic emissions dwarfed other changes, causing a 30% decrease in DSAOD over the reference period compared to only an 8% reduction from model configuration. Emission- and configuration-induced changes were comparable in India, and model configuration had a larger effect in Europe. Averaged over 0-70°N,
emission inventory and model configuration caused comparable reductions in simulated DSAOD; increased boreal latitude (50-60°N) biomass burning partially compensated for decreases in anthropogenic emissions. We emphasize that the changes in DSAOD stemming from model configuration led mainly to a systematic offset and had little impact on the anomalies considered in our main analysis.

The impact of anthropogenic emission inventory on total AOD and aerosol radiative forcing have been recently investigated
by Lund et al. (2023), who compare simulations using the two CEDS inventories analyzed here, as well as emissions from ECLIPSE v6. Although they do not assess the impacts of biomass burning emissions and they look at total rather than dust-

subtracted AOD, their findings are generally consistent with those presented here. In particular they identify a reduction in AOD and improved agreement with observational trends over 2005-2017, dominated by reductions in emissions over East Asia.

There exist substantial differences in biomass burning emissions between different inventories (Pan et al., 2020). We have used GFED v4.1s, which is among the low-emission datasets; it is possible that a different biomass burning inventory may have been more accurate. However, since in three of our four regions the MAM biomass burning emissions are small compared to anthropogenic emissions, this choice is unlikely to affect our results. If an underestimation of biomass burning emissions or their variability did impact our results, it would likely be in the Northern Hemisphere domain. Not only does the Northern

Hemisphere as a whole contains a higher proportion of biomass burning to anthropogenic emissions, but emissions are poorly constrained in the boreal latitudes due in part to a scarcity of observational data (Pan et al., 2020).

    In East China, discrepancy between observed and simulated anomalies may also be due in part to processes that were not captured in the models. It has been well documented that a combination of stagnant meteorological conditions, including an unusually weak East Asian winter monsoon, and a highly polluted background resulted in haze production when emissions

decreased (Gao et al., 2022; Chang et al., 2020; Huang et al., 2021; Kong et al., 2023; Le et al., 2020; Li et al., 2021; Shi and Brasseur, 2020; Wang et al., 2020; Xu et al., 2020; Zhao et al., 2022). Similar local enhancements in pollution levels were identified in a number of polluted regions around the globe (Balamurugan et al., 2021; Gaubert et al., 2021; Sicard et al., 2020; Venter et al., 2020). These enhancements occur because, in $NO_x$-saturated regions, the reduction of $NO_x$ emissions (largely due to decreased vehicle traffic) led to increased ozone production, which in turn increased the oxidative capacity of the atmosphere

and resulted in enhanced secondary aerosol formation (Kroll et al., 2020). These unusual meteorological conditions, and the resulting complex and nonlinear chemical reactions, are not well represented by the CovidMIP models. Indeed, of the models considered here, only MRI-ESM2-0 includes an interactive ozone scheme; the other models used a prescribed perturbation (Lamboll et al., 2021). Disagreement between the observed and simulated responses is therefore unsurprising. Conducting a similar analysis in models that simulate more complex chemistry would be a valuable direction for follow-up analysis, and the

inclusion of a prognostic ozone scheme will be critical for representing the climate and air quality effects of future emission reductions in polluted regions.

    Interpretation of the results for East China is further complicated by the presence of spatial structure in both the magnitude and tendency of aerosol emissions in this region, with urban areas characterized by high baseline emissions but negative trends whereas rural areas have lower emissions with positive trends. Modest differences in the relative magnitudes of these

565 components of the signal could likely have substantial impacts on the area-averaged mean responses derived from different datasets.

    We have assessed aerosol changes averaged over fairly large geographic regions, and this decision will influence our results. Selecting a smaller box, centred over the most populated urban areas, would likely yield a larger COVID-19 signal. On the other hand, the selection of a larger domain allows us to average over different potential trajectories, so we do not need to

570 worry about signals appearing and disappearing by being advected in or out of our analysis regions. Furthermore, while local

AOD changes are relevant when studying air quality, the regionally-averaged aerosol response to an emission reduction is more relevant for understanding potential climate impacts.

This analysis assesses the capability of models to realistically simulate the *net* effect of emission changes on aerosol optical depth. As described in Section 2.1, the focus on AOD folds in the effects of many different processes. In order to help disentangle the different drivers of model performance, future analyses would benefit from studies of aerosol concentrations, which are an intermediate stepping point between emission and optical depth. Additional insight could be gained through an investigation of individual aerosol species; sulfate, for example, would exhibit a stronger COVID-19 signal than total AOD does. We have not conducted such an analysis due to the limited availability of speciated data amongst the CovidMIP models.

## 6.2 Comparison to Previous Studies

We have taken the approach of applying a single consistent framework to all four analysis regions. There is also much to be learned from bespoke analyses of the observed changes in each individual region, and indeed numerous such analyses have been conducted.

In India, AOD reductions have been reported by Acharya et al. (2021); Gouda et al. (2022); Ranjan et al. (2020); Rani and Kumar (2022); Smith et al. (2022), and Wei et al. (2022). The magnitude of reduction has been found to vary regionally and with the period of 2020 considered, and for relative changes, on the choice of baseline; in general it is reported to be on the order of 10-15% when averaged over India as a whole. These reductions are comparable to those we find from MODIS and MISR (-13% and -14% respectively) but somewhat smaller than those returned by CALIOP and ACROS-C (-24% and -21%). We emphasize that, as highlighted above and also by Smith et al. (2022) and Wei et al. (2022), a substantial portion of the observed reduction is due to anomalously low dust optical depth.

In China, the measured AOD response depends sensitively on the choice of reference period, since the trend in aerosol emissions reversed in or around 2013 (Zheng et al., 2018). When underlying trends are accounted for, the 2020 AOD anomalies are not significant, although species including $NO_2$, $SO_2$, and $SO_4$ do exhibit substantial changes (Diamond and Wood, 2020; Field et al., 2021; Smith et al., 2022). When trends are not accounted for, AOD reductions in Central Eastern China range from -14% to -30% depending on the choice of reference period; compared to the 2016-2019 mean (similar to our 2015-2019 reference period), AOD in this region was reduced 14-17% (Field et al., 2021). AOD increased in South China (Field et al., 2021; Diamond and Wood, 2020; Acharya et al., 2021), possibly as a result of biomass burning in Thailand, Myanmar, and Laos (Field et al., 2021). In comparison, we find an observed decrease of 4-13% relative to the 2015-2019 mean, when averaging over a domain that contains both these areas of positive and negative anomalies.

In Europe, Smith et al. (2022) report detectable changes in the summer and autumn of 2020, but not in the spring which is our focus in this work. Springtime changes may be detectable when smaller averaging boxes are used: Ibrahim et al. (2021) report an AOD reduction in Central Europe but an increase around the edges of the domain, which when averaged together would be consistent with our non-detection. It has been demonstrated that this reduction is driven by an increase in the number of moderately-low AOD hours, as opposed to a few extremely low hours (van Heerwaarden et al., 2021).

As highlighted in the introduction, model-based studies have predominantly been focused on the climate response to aerosol emission reductions, rather than the AOD change itself. However, they generally report modest decreases in zonal average AOD over ∼30°N in the spring, particularly March, of 2020 (Fasullo et al., 2021; Fiedler et al., 2021; Forster et al., 2020; Fyfe et al., 2021; Gettelman et al., 2021; Jones et al., 2021; Weber et al., 2020). In many cases this reduction is only detectable in the ensemble mean. We find similar behaviour in the CovidMIP models (not shown); as expected, the observed signal is less clear.

## 7    Conclusions

The reduction in aerosol emissions associated with the COVID-19 pandemic provided a unique opportunity for Earth System Model evaluation. We have investigated changes in total and dust-subtracted aerosol optical depth during MAM 2020 in order to assess the AOD sensitivity of CMIP6-class Earth System Models. The impacts of internal variability, background emissions levels, and simulated perturbation size on our results have been assessed with a series of sensitivity tests.

Despite the dramatic economic and lifestyle changes associated with the COVID-19 lockdowns, a statistically significant reduction in observed aerosol optical depth is only identified over India. CovidMIP models reproduce the observed responses reasonably well over India and Europe but appear to over-estimate the magnitude of response in East China and the Northern Hemisphere (0°N-70°N). We demonstrate that this discrepancy can be partially resolved in CanAM through the use of an updated emissions inventory; investigating whether the other models would show similar improvement is an avenue for further study. The substantial uncertainty in remotely-sensed observations of AOD precludes a detailed assessment of the relative biases in different models. As such, this analysis motivates future research into the drivers of the systematic biases in satellite retrievals of aerosol fields, particularly in the context of monitoring future emission reductions which are expected to take place over the coming decades.

AOD sensitivity is critical for the prediction of short-term climate impacts to emission reductions, and our results will provide important context for interpreting the simulated climate impacts of proposed mitigation pathways of both aerosols and co-emitted greenhouse gases.

*Data availability.* This analysis utilized data from the CovidMIP project, available through the Earth System Grid Federation (https:// esgf-node.llnl.gov/search/cmip6/); the Moderate Resolution Imaging Spectroradiometer (MODIS; Platnick et al., 2017), hosted by NASA (https://ladsweb.modaps.eosdis.nasa.gov/missions-and-measurements/products/MOD08_M3 and /MYD08_M3; the Multi-angle Spectral Radiometer (MISR; Diner et al., 1998), hosted by NASA (https://asdc.larc.nasa.gov/project/MISR/MIL3MAEN_4); the Cloud-Aerosol Lidar with Orthogonal Polarization (CALIOP; Winker et al., 2009), hosted by NASA (https://asdc.larc.nasa.gov/project/CALIPSO/, product names CAL_LID_L3_Tropospheric_APro_AllSky-Standard-V4-20_V4-20 and CAL_LID_L3_Tropospheric_APro_CloudFree-Standard-V4-20_ V4-20); the ModIs Dust AeroSol (MIDAS) project (Gkikas et al., 2021), available on Zenodo (https://zenodo.org/record/4244106#.ZAfI-nbMKMo, data for 2018-2020 available from Antonis Gkikas by request), and the ACROS Global Dust Climatology (Song et al., 2021), hosted by

UMBC (https://acros.umbc.edu/data-and-models/decadal-global-dust-aod-database/). We thank these organizations for making their data publicly available.

*Author contributions.* RD led the study design, analysis, and writing, with support and supervision from AM, NG, and KvS. AG, QS, and ZZ provided data and comments on the manuscript.

*Competing interests.* The authors declare no competing interests.

*Acknowledgements.* This work was supported by the Natural Sciences and Engineering Research Council of Canada (NSERC; Grant RGPIN-2017-04043). AG has been supported by the Hellenic Foundation for Research and Innovation (HFRI) under the "2nd Call for HFRI Research Projects to support Post-Doctoral Researchers" (project ATLANTAS; project no. 544). We thank Slava Kharin for producing SST and SIC forcing fields for our CanAM simulations, David Winker for advice on the sources of uncertainty in CALIOP L3 datasets, and David Plummer for comments on the manuscript. We also thank the anonymous reviewers for their suggestions which improved this manuscript.

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
