# Peer review of "How well do Earth System Models reproduce the observed aerosol response to rapid emission reductions? A COVID-19 case study"

_EGUsphere, 2023_

## Author Comment (AC1)

**Reviewer 1**

In this study, the authors did lots of work to estimate AOD changes during spring 2020, based on satellite remote sensing products, and to evaluate AOD responses to emission reductions in several CovidMIP models. Unfortunately, the different satellite instruments gave such a large spread in AOD changes, even with dust (as one of the natural species) excluded in the retrievals. Strong regional dependence of the robustness of observational estimates and model performance is found, but drivers behind this are unclear. The analysis of CovidMIP models does not add much to the literature, beyond the original CovidMIP paper (Jones et al., 2021) and other published studies. These are the major concerns leading to my hesitation to recommend the current manuscript for publication. The CanESM5 sensitivity tests are more interesting and potentially revealing. The novelty and science significance of this paper may be increased by focusing more on in-depth analysis of the sensitivity experiments on the roles of meteorological factors and possibly microphysical processes driving the response of aerosols to emission reductions in spring 2020.

We thank the reviewer for their comments and suggestions on our manuscript, and hope that the revisions presented here will satisfactorily address their concerns. We note that the title of this manuscript has been updated to emphasize the focus on using COVID-19 observations to evaluate ESMs, rather than on studying the response to COVID-19 in and of itself.

To address the comments presented above:

- Contribution to the literature:
    - We believe that this work fills a gap in the existing literature on aerosol changes during COVID-19. The majority of existing studies focus on observations or simulations, but not both. Those that do compare observed and simulated responses generally use the simulations to better understand the observed anomalies, e.g. by predicting control conditions in order to determine how much of the observed anomaly can be attributed to anthropogenic emission reductions. To the best of our knowledge, no studies have yet used the observed COVID-19 response to evaluate model simulations of an equivalent reduction in emissions.
    - Although the original CovidMIP paper (Jones et al. 2021) does present a first look at simulated AOD changes during COVID-19, our work extends on theirs in several ways. Perhaps most importantly, Jones et al. (2021) looked exclusively at the simulated response, and did not include any comparison with observations. Furthermore, they present AOD anomalies without any investigation into the drivers of these changes, as their main foci were (a) the presentation of the CovidMIP experiment itself, and (b) the radiative/climatic effect of the combined aerosol and GHG emission reductions. We have added a paragraph to the introduction discussing their work and our extensions on it (see more detailed comment below).

- Drivers of regional changes:
    - We agree with the reviewer's comment that both observational estimates and model performance vary from region to region, and acknowledge that our discussion of these drivers may have been unclear. We have removed lines 451-463 of the original discussion, which discussed differences between the observed datasets in the Northern

Hemisphere, as it confused the overall message. We have also elaborated on the differences in simulated dust-subtracted AOD anomalies in the Northern Hemisphere: we have added the sentence, *"The spatial origins of these overestimations differ: in MRI-ESM2-0, the Northern Hemisphere-averaged anomaly is due almost entirely to the strong negative anomaly over Asia; in CanESM5, ensemble-median anomalies are negative throughout the entire region (Supplementary Figure S8)."* (lines 353-356), and clarified our discussion on the potential impacts of differing trends between the observed and simulated datasets (lines 357-365). When combined with the other updates to the manuscript, we hope that the existing discussion on the regional differences in both observed response and model performance will now be more clear.

- CanESM5 sensitivity tests:
  - We are glad that the reviewer finds these sensitivity tests interesting.
    We have edited the abstract to emphasize importance of these sensitivity tests to our analysis: line 10, *"we systematically assess"* to *"we conduct a series of sensitivity tests to systematically assess"*
    In addition, we have highlighted the potential of conducting similar sensitivity tests in the other CovidMIP models to determine how representative the CanAM results were (line 400/566); such analysis is outside the scope of this work, but would be illuminating. It could also be interesting to conduct a similar analysis in an air quality model such as GEM-MACH, which has the capacity to simulate gas-phase chemistry and more detailed aerosol processes; however, such a test would primarily be interesting in the context of studying COVID-19 itself, and would not aid in the present goal of assessing the ability of Earth System Models to simulate a COVID-19-like emission reduction.
    Finally, we have run an ensemble of simulations (CanAM-old-emis) to investigate the relative contributions of emission inventory and model configuration on simulated AOD. The results of this analysis are included in the discussion and explored in more detail in Supplementary Material S3.

Below are a few more specific comments:

- The abstract lacks quantitative results either from the observational estimates or model analyses.
  We have updated the abstract to clarify that our statements are grounded in quantitative statistical tests: line 6 now reads, *"…most regions do not exhibit **statistically significant changes…"*** These statistical tests form the basis of our analysis, and are the main results upon which we report.

- Line 4 (and several other places): strictly aerosol optical depth rather than aerosol burden is estimated in this study, which should be made clear.
  We acknowledge that the original manuscript incorrectly referred to aerosol burden in a number of places rather than AOD, and we thank the reviewer for pointing this fact out. We have clarified the text both to emphasize that we are investigating AOD, not burden, and to elaborate

on our reasons for doing so.

- Line 39-41: This statement is inaccurate. Jones et al. (2021) did specifically compare regional AOD changes among the participating Earth system models.
  While we believe that our original statement was correct, we understand that it lacked sufficient detail and could be misinterpreted.  We have added a more detailed description of the Jones et al. study to the Introduction, lines 71-76:
  *"The models used in this work are taken from the COVID-19 Model Intercomparison Project (CovidMIP; Jones et al., 2021), which was developed to investigate the effects of a COVID-19-like reduction in aerosol and greenhouse gas emissions. Although Jones et al. (2021) present an initial analysis of changes in aerosol optical depth, their primary foci were the radiative and climatic responses to the COVID-19 perturbation, and the drivers of the simulated aerosol changes were not investigated. Our analysis provides the first detailed investigation of aerosol changes in the CovidMIP models, as well as the first comparison between observed and CovidMIP-simulated changes."*

- Line 46: Please be more specific about what kinds of observed changes being used for model evolution purposes. Global or regional climate models use many different observational data for the evaluation purpose.
  We have updated the sentence to read, *"No studies have yet leveraged the observed aerosol response to the COVID-19 lockdowns for model evaluation purposes."*

- Line 61-64: depending on the purpose of obtaining aerosol concentrations, it can be a big problem of using a column optical property (AOD) as a proxy for aerosol concentration or aerosol burden mentioned in the first science question.
  Thank you for raising this concern. In the original text, these caveats were presented in the discussion, but we agree that they should have been presented in the introduction. In fact, there are reasons that the AOD is intrinsically interesting, and not merely an imperfect proxy for aerosol burden. We have updated the text to clarify our motivation for studying AOD, and to remove references to aerosol concentration/burden except as a possible avenue for future research.
  The text now reads, *"Both the air quality and climate impacts of a reduction in emissions depend on the response of atmospheric aerosol concentrations to emission changes, which is in general a complex and nonlinear dependence (Szope et al., 2021; Kroll et al., 2020).  The climate effect further depends on  the resulting changes in extinction, which can be quantified in terms of aerosol optical depth. In this work we consider changes in AOD, rather than concentration, since it is readily available from both model simulations and remotely sensed observations."*

- Line 78 (and section 2.1): It is too vague and generic to name meteorological conditions as one of the determining factors of AOD. In addition to the emissions, one should at least speak to the transport and sink terms of atmospheric aerosols such as dry and wet deposition in aerosol budget equation, although the detailed aerosol chemical and microphysical processes are sometimes even more important, depending on the aerosol types.
  We have expanded Section 2.1 to more clearly describe the processes that drive AOD variability.

We now discuss separately the factors that determine aerosol burden (emissions, secondary production, transport, and the effects of meteorology on lifetime via wet and dry deposition rates) and those that determine the AOD that results from a given burden (characteristics of the aerosol such as morphology and refractive index, and microphysical processes such as hygroscopic growth).

- Table 1: There might be too few models. Are they outliers among the 12 CovidMIP models?
  We thank the reviewer for their concern on this point. We would of course have preferred to include more models in our analysis, but were limited by data availability. However, the AOD anomalies simulated by these models span the range of AOD anomalies shown in the original CovidMIP paper, as well as the range of anomalies in downwards SW radiation flux, global surface air temperature, and global precipitation response. As such we feel that it is a representative sample.
  We have updated the text to include (lines 184-185), *"These six models, summarized in Table 1, sample the range of global AOD anomalies and climatic responses simulated by the full CovidMIP suite (Jones et al., 2021)."*

- Line 228-231: How was the first assumption tested?
  Section 4 (lines 245-289) has been revised to include an explanation of how this assumption was tested, and slightly reordered for overall clarity and readability. The text directly addressing this comment reads,
  *"We assess the first of these assumptions by comparing observed and simulated variability over the reference period in our regions of interest. We calculate the variance of the region-mean AOD field over the reference period for each simulated ensemble member, the MMEc, and the observational datasets, and compare the spread in these estimates of variance. Although individual models may over- or underestimate the interannual variability in some regions, we cannot reject the null hypothesis that the MMEc and observations have the same interannual variability, based on a two-sided Welch's t-test. The only exception is for India, where the MMEc overestimates the variability in total and dust-subtracted aerosol optical depth; as a result, estimates of observational detectability will be conservative (i.e., anomalies are less likely to be found to be statistically significant). In a similar analysis of the variability over a longer baseline (2007-2019), using a subset of models for which these data were available, the total-AOD variability of the MMEc is consistent with that of the observations in all four regions."*

- Line 284: This statement about contribution of dust to total AOD is inaccurate and can be misleading. It highly depends on season and region. Globally, dust contributes to less than 25% of annual total AOD.
  We thank the reviewer for this correction. We have conducted further analysis and, while the dust indeed does not dominate the total AOD, it does dominate the variability in our regions of interest (see also Gkikas et al. (2022), Fig. 10; ). The text has been updated accordingly. Lines 324-326 now read, *"We next investigate the AOD signal when the contribution from mineral dust has been removed. In our regions of interest, the variability in total aerosol optical depth is dominated by the variability in mineral dust, which was not directly impacted by COVID-19 lockdowns."*

References:

Gkikas, A., Proestakis, E., Amiridis, V., Kazadzis, S., Di Tomaso, E., Marinou, E., Hatzianastassiou, N., Kok, J., and Garcia-Pando, C. P.: Quantification of the dust optical depth across spatiotemporal scales with the MIDAS global dataset (2003–2017), ACP, 22, https://doi.org/10.5194/acp-22-3553-2022.

**Reviewer 2**

The manuscript "How well do Earth System Models reproduce observed aerosol changes during the Spring 2020 COVID-19 lockdowns?" use the COVID lockdown and the following emission reduction for model evaluation. Modelled changes in aerosol optical depth (AOD) due to COVID restrictions and satellite retrieved AOD in March, April, May (MAM) 2020 are compared. The Earth System Models and observations show consistent results in Europe and India, where India is the only region considered with a significant reduction in AOD in MAM 2020 in the observations. In China and Northern Hemisphere as a whole, the modelled reduction in AOD is overestimated. Using one model, a systematic assessment of the influence of meteorology, baseline emissions, size of COVID emission reductions are done. The spread in the observations of AOD is a limiting factor of further constraining the models.

We thank the reviewer for their very helpful comments on this manuscript. Our responses to their specific suggestions are provided below. In addition, we wish to note that the title of this manuscript has been updated to emphasize the focus on using COVID-19 observations to evaluate ESMs, rather than on studying COVID-19 in and of itself.

The manuscript is well structured and presented, and I have only a few comments.

- The uncertainties in the satellite AOD products precludes a further constraint on the models responses to emission reduction. As this method outlined here, also can be used to evaluate response to future emission reductions, a bit more on future direction of satellite AOD evaluation would have been good.
  This is an excellent point. We have added to the conclusion (line 582-585), *"The substantial uncertainty in remotely-sensed observations of AOD precludes a detailed assessment of the relative biases in different models. As such, this analysis motivates future research into the drivers of the systematic biases in satellite retrievals of aerosol fields, particularly in the context of monitoring future emission reductions which are expected to take place over the coming decades."*

- L4: "observed regional aerosol burdens during" It is not aerosol burden that is assessed, but AOD (as a proxy for aerosol burden).
  We acknowledge that the original manuscript incorrectly referred to aerosol burden in a number of places rather than AOD, and we thank the reviewer for pointing this fact out. As detailed in our responses to Reviewer 1's comments on Lines 4 and 61-64, we have clarified the text both to emphasize that we are investigating AOD, not burden, and to elaborate on our reasons for doing so. References to aerosol concentration and burden have been removed from the text except as a possible avenue for future study.

- Consider swapping section 2.3 and 2.2?
  We have decided to keep the sections in their current order, but have expanded the text to clarify our logic for doing so.
  Lines 87-89 now read, *"We begin by highlighting the major considerations that need to be addressed in an analysis of this type: sources of AOD variability; differences that are expected to arise between simulated and observed AOD fields, no matter the quality of the atmospheric model or satellite retrieval; and finally, the impacts of observational uncertainty."*

We have also added the following sentence to the end of Section 2.1 (lines 111-113): *"In the following sections, we describe first the differences that would be expected even if both models and observations were perfectly accurate, and then the impacts of observational uncertainty."*

- Table 1: The mineral dust column can be misread as if models include mineral dust or not in the simulations. Maybe replace "Mineral dust?" by "od550dust" or "Mineral dust output".
  Thank you for pointing out that this is unclear. We have updated the column heading to read, *"Published od550dust?"* and in the caption specify that this indicates whether od550dust was available on ESGF.

- Section 3.2: Could be useful with a table of the satellite AOD products.
  Thank you for the suggestion. We have added a table summarizing the major features of the satellite AOD products.

- ACROS-C is used in Figure 1, but not mentioned in section 3.2.1.
  Thank you for catching this omission. ACROS-C has been added to Section 3.2.1.

- L253: "the Northern Hemisphere as a whole" From figure 1 and text elsewhere, the "as a whole" is not entirely correct as it is 0N-70N. Replace "as a whole" with (0N-70N).
  We thank the reviewer for pointing out this inconsistency. The text has been updated accordingly. In some cases, the phrase "as a whole" has been retained, but the latitudinal range has been added, e.g. *"When the Northern Hemisphere (0-70N) is considered as a whole, …"* to avoid making it sound as though we consider the Northern Hemisphere to be somehow separate from the other regions which are contained within it.

- Figure 1 (and 2 and 3) contain a lot of information. It could be useful to add more information to the legend, maybe first present what is included in the timeseries (the six models and the observations with symbol and black line). Then, as a separate box or just below, the 2020 values (Square: control, diamond: covid pert, MMEc). For MMEc maybe only show the square and not the line, as I was looking at the time series when I first looked at the plot. See also if filled, black outline, opaque/semi-transparent can be indicated inside the figure as well. I am not able to see if the results are plotted opaque or semi-transparent. Possible to use filled or not filled symbols instead?
  We thank the reviewer for their recommendations, and in particular for highlighting the challenge in differentiating between opaque and semi-transparent markers. We have made the following changes:
  - The statistical significance of the 2020 anomalies is now indicated by filled vs open markers, as opposed to opaque vs semi-transparent.
  - Simulated ensembles that are consistent with the observed ensemble are indicated by a black centre dot, rather than a black outline on the marker.
  - The legend has been split into two, with the second legend summarizing the formatting conventions of the 2020 points.

- Figure caption: Delete "horizontal offset for visual clarity" Already mentioned that the right side of the panel was for 2020 and "2020 values" are the titles of the subpanels.
  Updated.

- L313: It is hard by eye to see the difference in the trend between observations and models for the reference period (2015-2019).
  The language in this section has been softened to avoid claiming the existence of a trend; we agree with the reviewer that a difference is not clearly visible, especially given the variability in the observations. We do still include some discussion around the potential impacts of a difference in trends, since the raw data do suggest that such a difference *might* exist, and as discussed in Section 4 our use of the MMEc assumes that the observations and simulations have similar trends. As such it seems important to address the possibility. The original text from lines 312-319 has been replaced with, *"Given the short reference period and substantial interannual variability of the observations, it is challenging to identify whether the simulated trends are representative of those observed. In East China there is some indication that the models may simulate marginally more negative trends than the observations (more visible in the raw data shown in Supplementary Figure S6, and when the 2020 anomaly is not included in the timeseries), which could imply that the MMEc may inadequately represent the range of plausible "control observations."' However, this discrepancy -- if it exists -- does not appear sufficient to explain the absence of a statistically significant anomaly in the observations. Visual inspection suggests that the observed 2020 anomalies are consistent with AOD excursions measured over the preceding 5 years, even taking any potential trends into account, whereas the simulations show an obvious decrease in 2020. This behaviour is particularly clear when considering the raw data shown in Supplementary Figure S6."*

- L359-362: This was a bit unclear. Just note that the MMEc is as in Figure 2.
  Thank you for the suggestion. This paragraph has been removed, and the necessary information added to the caption of Figure 3. In fact, the MMEc is different in Figures 2 and 3 (it is drawn from only the models included in the figure) but as it turns out, the choice of MMEc does not change whether the observed anomaly is statistically significant; we agree that this was unclear in the original text.

---

## Author Response (AR2)

We thank the 3 anonymous reviewers for their helpful comments on our manuscript. Their comments, and our corresponding edits and responses, are copied below.

**Reviewer 1**

I assume MAM stands for March-April-May. It is not defined in the paper.

Thank you for catching this omission. Line 254, the first time we use the acronym, now reads, *"Throughout, anomalies are calculated relative to the 2015-2019 March-April-May (MAM) mean."*

CanESM5 is noted as "CanESM50" in the plot legend of both Fig. 1 and Fig. 2. Is that a mistake?

We have clarified both the figures and text to be more explicit about model versioning. Both now refer to the model as "CanESM5.0" throughout the manuscript, except for occasional references to CanESM5.1 or CanESM5 in general as appropriate.

In the caption of Figure 4, please make a note of the model name used for the sensitivity experiments shown in this figure.

Thank you for noting this omission. The caption now reads, *"Effects of varying the magnitude of the simulated COVID-19 perturbation in CanAM5.1."*

**Reviewer 2**

L6: "forced with COVID-19-like reductions in aerosol and greenhouse gas emissions" The models were actually forced by greenhouse gas concentrations and aerosol and aerosol precursor emissions. Possible suggestion: "forced with COVID-19-like reductions in aerosols and greenhouse gases".

We thank the reviewer for catching this oversight. We have corrected both the abstract (line 2) and main text (lines 79, 169) as suggested. The only place where we still refer to "reductions in aerosol and greenhouse gas emissions" is in the description of the original CovidMIP methodology, which did estimate reductions in the emissions of both aerosols and greenhouse gases; as the reviewer points out, the GHG emissions were then converted to concentrations for use in model simulations.

**Reviewer 3**

Page 3, L60: how was COVID different than long term changes? You mention co-emission of species. Did COVID reductions change that? My guess is probably: energy emissions were the same, but transport emissions were reduced. Maybe discuss this?

Thank you for the excellent question. We have expanded on our reasons for being interested in a rapid change (lines 61-68):

*There are both practical and scientific motivations for studying a rapid emission reduction. On the practical front, a short-but-strong signal is easier to disentangle from other sources of variability; we have continuous satellite observations that cover the entire study period; and the period is short enough*

*the instrument drift is unlikely to be a concern. Scientifically, model simulations indicate that the presence and severity of potential climate penalties, including changes in mean and extreme temperatures and precipitation, may be proportional to the rate at which emissions are reduced (Acosta Navarro et al., 2017; Hienola et al., 2018; Samset et al., 2020; Shindell and Smith, 2019; Shindell et al., 2012; Sillmann et al., 2013). Although we do not investigate the climate response to COVID-19 in this work, understanding the aerosol response itself is an important first step.*

Page 4, L115: You might want to note explicitly that satellite retrieval error and model error are two other significant error sources.

We agree with the reviewer that satellite retrieval error and model error are important sources of error; these are discussed in the following sections. This section specifically discusses the factors that would contribute to differences between observed and simulated AOD signals even in the absence of those key sources of uncertainty. We have revised the text to clarify the intent of this section.

Lines 94 to 96 now read, *"We begin by highlighting the major considerations that need to be addressed in an analysis of this type: sources of AOD variability; factors that contribute to discrepancies between simulated and observed AOD fields, no matter the quality of the atmospheric model or satellite retrieval; and finally, the impacts of observational uncertainty."*

Section 2.2 has been renamed, *"Differences between observed and simulated AOD in the absence of model error or observational uncertainty"*

Lines 122 to 129 now read, *"Even given a hypothetical model that perfectly simulated atmospheric aerosol processes, and perfectly accurate satellite retrievals, differences would still arise between the observed and simulated AOD fields. These differences can be grouped into three main categories. First, a freely running model would produce a different realization of meteorological conditions than occurred in the real world, and so aerosols would be subject to different emission, transport, and depositional processes. Second, any errors in the model inputs (e.g. in the size of perturbation applied to represent COVID-19) would translate into biased simulations. Finally, the simulated and observed AODs would be recorded with different spatiotemporal sampling. Before any differences between the observed and simulated responses to COVID-19 can be attributed to model biases, then, these factors must be accounted for. (We discuss the role of observational uncertainty separately, in Section 2.3.)"*

Page 6, L170: How accurate was the 2 year blip assumption? Can you compare it to mobility data until now? Since it sounds like you are going to conclude emissions matters, maybe you can show the assumptions versus actual observations/inventories?

Assessing the accuracy of the 2-year blip assumptions is beyond the scope of this analysis, and up-to-date aerosol emission inventories, such as those from CEDS or ECLIPSE, are not yet available for 2020. Although regional assessments have been attempted by other researchers, a global evaluation of the scenario is not yet available (Forster et al. 2023). However, we assess the degree to which uncertainties or errors in the two-year blip assumptions would impact our results in Section 5.2.3.

Page 7, L186: Is Sigmond et al the reference for the CanESM5 AOD spread or do you need to show a figure?

Yes, Sigmond et al is the reference for the CanESM5.0 AOD spread; see in particular Section 5.1. This paper has now been published in GMD and the citations updated accordingly.

Page 10, L260: I don't think using the models to define observational anomalies is really appropriate. Why don't you use the pre-covid period and it's variability to define the mean and variability for the t-test?

We appreciate the reviewer's concern over this approach. To clarify, the anomalies themselves are calculated with respect to the mean over the 2015-2019 reference period. Unfortunately, using previous years' observations to determine the AOD expected in 2020 in the absence of the COVID-19 perturbation (an observational "control") is not practical. We have clarified the motivation for our approach in the manuscript; lines 261-267 now read,

*Defining detectability for the observations is more challenging, because there is no ``control observation'' from which to estimate the AOD that would have been measured in 2020 had COVID-19 not occurred. It would not be sufficient to use the mean and variance calculated from the reference period as a control: using the mean would neglect the impacts of underlying trends in the emissions, and a five-year reference period is too short to provide a robust estimate of the variance. Instead we borrow an approach from the field of detection and attribution (Eyring et al., 2021) and compare the ensemble of observed anomalies to a multimodel control ensemble (MMEc) constructed by randomly drawing an equal number of control simulation anomalies from each model.*

As described in lines 270-278, this approach is supported by our comparison between the observed and simulated data over the reference period, as there is not a statistically significant difference between the interannual variability of the MMEc and the observations.

Page 10, L280: are you doing this at each point or some global mean? I hope it is at each point: there is information in the pattern.

We are doing this analysis using region-mean values, for each of our 4 analysis regions. We agree that a global-mean result would not be meaningful! Spatially-resolved t-tests do not change our results, and we have added a sentence clarifying this fact to the manuscript: *"We present results for t-tests performed on region-mean values; using spatially-resolved comparisons adds little information and does not change our results."* (line 289-291).

For the reviewer's interest, we have copied a figure showing the results of our t-tests for the model CanAM-new-emis over the Northern Hemisphere domain. In each panel, regions in orange/red have p<0.05, and blue/purple have p>0.05. The top panel compares control and perturbed ensembles, so regions in red exhibit statistically significant differences between the ensembles. As in the region-mean results, significant anomalies are found over India and China, but not Europe. The middle and bottom panels compare the control and perturbed ensembles respectively against the observations. Here, purple indicates areas of good agreement between the observations and simulations (where there is not a statistically significant difference between the two). White indicates regions of missing data. The spatial patterns shown in these panels add little information to our analysis, especially considering the fact that by definition, 5% of the region will have p<0.05.

[Figure]

Page 11, L300: note that MAM = March - May?

We thank the reviewer for noting this omission. Line 254, the first time we use the acronym, now reads, *"Throughout, anomalies are calculated relative to the 2015-2019 March-April-May (MAM) mean."*

Page 11, L304: define "detectable"

We have replaced the word "detectable" with "statistically significant."

Page 14, L370: Please clarify this is the same model as the CanAM5 in the previous plots (you corrected it for this paper).

We have revised the text to be more explicit about model versioning. The preceding analysis used the coupled earth system model CanESM5.0, and the sensitivity tests described here use the atmospheric model CanAM5.1. CanAM5 is the atmospheric component of CanESM5; the upgrades from version 5.0 to 5.1, described in Sigmond et al. (2023), resolve the dust issues described earlier in the manuscript.

Page 15, L389: I don't agree, it is in line with one of the data sets, and that is unchanged from the regular emission version given the spread.

We acknowledge that by eye, the updated-inventory results appear similar to those with the original CMIP6 emissions. Our statement refers to the results of our t-test: with the original emissions, the separation between the observations and the perturbed ensemble was statistically significant, but with the updated emissions, the separation is not statistically significant. We have clarified the text to emphasized that our comparisons with the observations are grounded in statistics (and to clarify where our discussion is qualitative). Lines 399-403 now read:

*The effects of this reduction vary: in East China, the update is sufficient to bring the simulated COVID-19 signal into agreement with the observed anomaly **(i.e., the separation between observed and simulated***

*ensembles is no longer statistically significant),* whereas *in the Northern Hemisphere the simulated response is brought somewhat closer to the observations but the difference remains significant. In Europe the main effect of the update is to remove the simulated trend over the 2015-2019 reference period, bringing the simulations into qualitatively better agreement with the observations over this period…*

We have also clarified these comparisons earlier in the results section. The figure legends have been updated so that the black dot used to denote agreement between the simulated and observed ensembles is now labeled *"sim-obs dif. not stat. sig.,"* and the caption to Figure 1 now reads,

*"Black dots indicate simulated anomalies that are not significantly different from the ensemble of observed anomalies."*

Lines 322-324 have been updated to read,

*In East China, four of the six models exhibit a statistically significant COVID-19-perturbed anomaly, which in general appears to be over-estimated: in these models, there is a statistically significant between the observations and the perturbed ensemble, but not between the observations and the control.*

Page 15, L397: you just said Europe is improved with the new emissions. Be consistent or more precise.

We thank the reviewer for pointing out this discrepancy, and agree with their assessment. Lines 407 to 411 now read, *"These results suggest that in India, inferences drawn from the other CovidMIP models will likely not be affected by biases in the underlying baseline inventory as long as the applied perturbation is realistic. In the Northern Hemisphere, East China, and Europe, the apparent overestimation of the COVID-19 response identified in CanESM5.0, MIROC-ES2L, and MRI-ESM2-0 may have been partially caused by overestimates of the control emissions and thus of the absolute magnitude of the COVID-19 disruption."*

Page 16, L417: for the 2020 covid decrease. It does NOT improve earlier agreement.

We have clarified the text such that lines 44-52-453 now read, *"… the 2020 perturbed anomaly is in excellent agreement with the observations, …"* See also our more comprehensive revisions, discussed under your comment on Page 17 L43.

Page 17, L420: Again, I don't agree with this interpretation. What about anomalies in 2017 and 2018? Please be specific here.

This statement was intended to refer to specifically to the anomalies in 2020. We recognize that it was unclear, and have revised the text to be more explicit about our interpretation. In the process, we have substantially revised Section 5.2.2, and the paragraph referred to in this comment has largely disappeared. Please refer to the following comment for a description of the changes made.

Page 17, L431: but the inter-annual variability is not reproduced. the observations tend to have the same sign of inter-annual anomalies and the nudged model does not. Please explain how this is consistent?

We have substantially revised Section 5.2.2 to clarify the interpretation of the nudged results, in the context of differences in interannual variability between the different datasets. In particular, we explicitly address the higher variability of the nudged simulations (lines 425-429):

*In general the nudged simulations exhibit higher interannual variability than the free-running simulations do, although this variability generally falls within the envelope of the larger coupled ensemble. In Europe, the nudged ensemble exhibits substantially higher interannual variability than do either of the free-running ensembles or the observations, indicating that the model may underpredict the variability of and AOD sensitivity to temperature, winds, and/or humidity in this region.*

We address differences in the sign of the anomalies in Europe (lines 453-457):

*The pattern of positive and negative anomalies is consistent with, although substantially amplified relative to, observations from MODIS-Aqua; however, this pattern of variability is not reproduced in the two CALIOP-derived datasets. The simulated COVID-19 perturbation is small relative to this variability and, as in all previous ensembles, the control and perturbed 2020 anomalies are both consistent with the observed ensemble.*

We soften our description of the nudged results in India (lines 449-452):

*Because the observed interannual variability is reproduced less well in the nudged than the free-running simulations, detailed comparisons between the observed and simulated 2020 anomalies would not be robust; however, it is reassuring to note that the 2020 perturbed anomaly is in excellent agreement with the observations, and the results of our statistical comparisons remain unchanged.*

We discuss the variability of the different datasets when summarizing results for East China and the Northern Hemisphere (lines 439-441 and 458-460 respectively):

*In East China, both CanAM-new-emis-free and CanAM-new-emis-ndgd simulate lower interannual variability than do the observations; the pattern of variability is perhaps somewhat improved in the nudged ensemble.*

*When averaged over the entire Northern Hemisphere (0N-70N), neither the free-running nor nudged ensembles reproduce well the observed pattern of interannual variability. Overall, our results here are unaffected by nudging: the control ensemble is consistent with the observed anomalies, and the perturbed ensemble values are too negative.*

Finally, in lines 461-464 (the original focus of this comment) we have provided more support for our argument that improvements in the observations are necessary:

*In all regions, the separation between the control and perturbed CanAM-new-emis-ndgd ensembles is on the order of the spread in observational estimates. Furthermore, in two regions (India and Europe), the three observational datasets disagree on the pattern of positive and negative anomalies throughout the reference period. These findings taken together suggest that further observationally-based model evaluation may not be feasible given current observational uncertainties.*

Page 18, L465: in the previous section you implied that nudging did improve agreement (please explain or make consistent).

We hope that the revised Section 5.2.2 will address this concern, as we are now more explicit about the areas in which nudging did and did not improve the agreement between observations and simulations.

**References**

Eyring, V., N.P. Gillett, K.M. Achuta Rao, R. Barimalala, M. Barreiro Parrillo, N. Bellouin, C. Cassou, P.J. Durack, Y. Kosaka, S. McGregor, S. Min, O. Morgenstern, and Y. Sun, 2021: Human Influence on the Climate System. In Climate Change 2021: The Physical Science Basis. Contribution of Working Group I to the Sixth Assessment Report of the Intergovernmental Panel on Climate Change [Masson-Delmotte, V., P. Zhai, A. Pirani, S.L. Connors, C. Péan, S. Berger, N. Caud, Y. Chen, L. Goldfarb, M.I. Gomis, M. Huang, K. Leitzell, E. Lonnoy, J.B.R. Matthews, T.K. Maycock, T. Waterfield, O. Yelekçi, R. Yu, and B. Zhou (eds.)]. Cambridge University Press, Cambridge, United Kingdom and New York, NY, USA, pp. 423–552, doi: 10.1017/9781009157896.005.

Forster, P.M., C. J. Smith, T. Walsh, W.F. Lamb, R. Lamboll, M. Hauser, A. Ribes, D. Rosen, N.P. Gillett, M.D. Palmer, J. Rogelj, K. von Schuckmann, S.I. Seneviratne, B. Trewin, X. Zhang, M. Allen, R. Andrew, A. Birt, A. Borger, T. Boyer, J.A. Broersma, L. Cheng, F. Dentener, P. Friedlingstein, J.M. Gutiérrez, J. Gütschow, B. Hall, M. Ishii, S. Jenkins, X. Lan, J.-Y. Lee, C. Morice, C. Kadow, J. Kennedy, R. Killick, J.C. Minx, V. Naik, G.P. Peters, A. Pirani, J. Pongratz, C.-F. Schleussner, S. Szopa, P. Thorne, R. Rohde, M.R. Corradi, D. Schumacher, R. Vose, K. Zickfeld, V. Masson-Delmotte, and P. Zhai.: Indicators of Global Climate Change 2022: annual update of large-scale indicators of the state of the climate system and human influence, Eart System Science Data, 15, https://doi.org/10.5194/essd-15-2295-2023, 2023.

Sigmond, M., Anstey, J., Arora, V., Digby, R., Gillett, N., Kharin, V., Merryfield, W., Reader, C., Scinocca, J., Swart, N., Virgin, J., Abraham, C., Cole, J., Lambert, N., Lee, W.-S., Liang, Y., Malinina, E., Rieger, L., von Salzen, K., Seiler, C., Seinen, C., Shao, A., Sospedra- Alfonso, R., Wang, L., and Yang, D.: Improvements in the Canadian Earth System Model (CanESM) through Systematic Model Analysis: CanESM5.0 and CanESM5.1, Geosci. Model Dev., 16, https://doi.org/10.5194/gmd-16-6553-2023, 2023.

---

## Author Response (AR3)

**Editor Comments**

*Dear Authors,*

*Please improve the colour schemes used in your maps and charts as indicated by the Editorial office.*

*There are COVID-19 related studies that can be referred and discussed in the paper, e.g.,*

*Wei, L., Lu, Z., Wang, Y. et al. Black carbon-climate interactions regulate dust burdens over India revealed during COVID-19. Nat Commun 13, 1839 (2022). https://doi.org/10.1038/s41467-022-29468-1.*

*Xiaohong Liu*

**Author Response**

Dear Dr. Liu,

Thank you for the publication decision, and for these notes. We have made the following changes:

*Colour schemes:*

1. The colour scheme in Figures 1-3 has been updated to use a subset of the colourblind-friendly rainbow scheme by Paul Tol (https://personal.sron.nl/~pault/), and the results checked using the Coblis colour blind simulator.
2. The colours in Figures 4, S06, S11, and S12 have been updated to be compatible with the new choice of colours for CanAM5.0, and the results checked using the Coblis colour blind simulator.
3. The colours in Figure S01 are left unchanged, but the green line denoting monthly-mean quantities has been made thicker to distinguish it from the thinner orange line.

*Wei et al. (2022) publication:*

- Thank you for bringing this study to our attention. We have cited it in our discussion of the anomalously low dust over India in 2020 (lines 343-345):

  *This improvement is partially due to the fact that the observed dust optical depth was anomalously low in 2020, as seen in our results and as reported elsewhere (Smith et al., 2022; Wei et al., 2022).*

- We have also cited it in our comparison with other published results (lines 583-584):

  *In India, AOD reductions have been reported by Acharya et al. (2021); Gouda et al. (2022); Ranjan et al. (2020); Rani and Kumar (2022); Smith et al. (2022), and Wei et al. (2022).*

*Minor text change:*

- We have made the following minor modification to lines 516-518 to improve clarity (bold words added):

  *Models appear to overestimate the COVID-19 response in the Northern Hemisphere **generally** and in East China **specifically**, although our results suggest that a more accurate emissions inventory can reduce this discrepancy.*